# Single-cell genomics highlight MYC-associated metabolic activation and altered cell interactions in T-prolymphocytic leukemia progression

Linus Wahnschaffe [1,2,3,4], Dennis Jungherz[1,2,3,5], Tony A. Müller[1,2,3], Tea Pemovska [6], Alexander Pichler [6], Stéphanie Poulain[7,8], Edith Julia [9,10,11], Sanna Timonen [12,13,14], Martin Böttcher [15,16], Qu Jiang [1,5], David Beverungen[5], Julia Bischoff[5], Marek Franitza[17], Theodoros Georgomanolis [17], Kerstin Becker [17], Michael Hallek [1,2,3], Dimitrios Mougiakakos [15,16,18], Thorsten Zenz [19], Satu Mustjoki [12,13,20], Emmanuel Bachy [9,10,11], Charles Herbaux [21], Alexandra Schrader [1,2,3,9], Natali Pflug[1], Philipp B. Staber[6,22], Till Braun [1,23] & Marco Herling [1,2,3,5,23] ✉

T-prolymphocytic leukemia (T-PLL) typically presents with rapidly progressing tumor burden. However, 15–25% of cases are diagnosed at an indolent stage with asymptomatic and stable low-level blood lymphocytosis over up to 2-3 years before advancing to active-stage disease. To define the molecular changes underlying this transition, we perform single-cell RNA sequencing of 28 treatment-naïve samples including 11 longitudinally acquired indolent/active pairs, paralleled by longitudinal whole genome sequencing. This reveals both patient-specific lesions and common global alterations of gene expression. Strong upregulations of MYC-target gene signatures in active T-PLL samples associated with enhanced energy metabolism implicate acquired autonomy from energetic restrictions. Recurrent downregulation of genes of the T-cell-receptor signaling cascade and reduced interactions of the T-PLL cell with non-leukemic cell types further indicate progressive independence from regulatory survival signals and escape from micromilieu-mediated control. This single-cell and disease-stage resolved genomic analysis of T-PLL provides insights into shared mechanisms of tumor evolution, which have to prove their amenability as targetable lesions.

T-prolymphocytic leukemia (T-PLL) is a neoplasm of mature T cells[1,2]. Clinical presentation of the majority of T-PLL patients is characterized by aggressive disease kinetics, such as exponentially rising peripheral blood (PB) lymphocytosis and repressive bone marrow (BM) infiltration with consecutive hematopoietic insufficiency[2–4]. At this typical stage around first diagnosis, infiltrations of lymphoid and non-lymphoid tissues further contribute to the symptomatology and include splenomegaly, lymph-node enlargements, hepatomegaly, polymorphic skin involvement, and effusions[2–6].

Patients with newly diagnosed T-PLL continue to face a disappointing median overall survival of 24 months. Attributing to this persistently poor prognosis is an inherent resistance towards current

(chemo)therapeutic options[7,8]. Despite response rates of ≈90% following first-line therapy with the monoclonal anti-CD52 antibody alemtuzumab (+/− conventional cytostatics)[3,9,10], relapses are inevitable and sustained leukemic control can only be accomplished by a consolidating allogeneic stem cell transplantation in eligible patients[11,12]. Although remarkable improvements have been made over the last two decades in the genomic and functional characterization of T-PLL, disease-specific therapeutic strategies still lag behind the advancing biologic concepts[7,13,14].

Current models of the protracted evolution from a post-thymic pre-leukemic cell propose constitutive activation of *T-cell leukemia/lymphoma* family molecules through chromosome-14 rearrangements (i.e., *TCL1A;* or of *MTCP1* at chromosome X) as the key initiating lesions[2,13,15–18]. Known oncogenic functions of TCL1A include augmentation of pro-survival kinases such as AKT or ERK resulting in amplification of T-cell receptor (TCR) stimuli[15,19–24]. The relevance of TCL1A-driven pathways in overt or progressing T-PLL has not been thoroughly interrogated.

The most frequent 'secondary' lesions of T-PLL, likely cooperating with TCL1A, comprise a perturbed DNA damage response caused by mono-allelically deleted and/or mutated *ATM* upstream of a genomically intact *TP53*[13,25,26], resulting in arrested repression of p53 and its pivotal functions[13,27]. *MYC* copy-number gains[13,17,28] and activating mutations of *JAK/STAT* genes[25,29,30] are further recurrent aberrations, but without resolved early- vs late-stage relevance in leukemic development.

Naturally, these data on the molecular make-up of T-PLL almost exclusively derive from cells of the active disease stage. However, a sizable subset of ≈15–25% of T-PLL patients are (often coincidentally) diagnosed with an indolent disease[2,3,31]. These patients are mostly asymptomatic, maintaining a stable disease with low-to-moderate tumor load and kinetics. As they virtually all advance to a rapidly progressing T-PLL after ≈2–3 years, it has been accepted that there is no relevant subset of a long-standing indolent T-PLL (hence abandonment of the term T-CLL)[31]. It is not fully resolved whether rare cases of longer indolent courses and TCL1A-negative status represent variants of regional prevalence or legitimate T-PLL[32].

Recognizing this indolent stage rather as an early phase towards an inevitable aggressive disease has implications on the initial treatment strategies (e.g., watch-and-wait vs early intervention), particularly to be (re)considered in light of available therapies. We proposed clinical cutoffs to discern indolent-stage T-PLL from active T-PLL, e.g., PB lymphocytes exceeding 30 G/l or their doubling time being ≤6 months, or becoming symptomatic, but these criteria were mainly intended as treatment triggers[2].

Systematic investigations of the pathobiology of indolent-stage T-PLL would provide informative access to early disease stages and by that overcome the limitations of chronological experimental models, such as *Lck^pr-hTCL1A^tg* mice[13,33]. Consequently, molecular analyses of indolent-phase T-PLL in comparison to active T-PLL (ideally within the same patient) harbor the potential to deduce inherent phase-specific vulnerabilities (and resistances) to guide future therapeutic approaches.

Here, we employed single-cell-resolved transcriptomic analyses of 11 longitudinally-paired and 6 single time-point derived T-PLL samples to unveil the molecular changes underlying the evolution from indolent to active T-PLL. We identified both patient-specific mechanisms of tumor evolution and common patterns of disease progression. Our findings suggest a key role of energy metabolism in the transition from indolent to active disease, accompanied by a strong upregulation of *MYC* gene expression signatures. Furthermore, reduced cell-cell interactions in active T-PLL implicate increased independence from milieu interactions. This was paralleled by decreased signatures from exogenous growth stimuli such as via TCR and TNFα/NFκB/MAPK signaling pathways, which we established as essential drivers of early

T-PLL. Additional longitudinal whole genome sequencing (WGS) further pointed towards an outgrowth of ATM-deficient T-PLL cell clones and progressive accumulation of *MYC* copy number alterations (CNAs).

## Results

### A single-cell transcriptome atlas of sequentially acquired samples during T-PLL evolution

To investigate the mechanisms underlying the transformation from indolent to active T-PLL we performed single-cell RNA sequencing on 11 longitudinally sampled indolent/active pairs, each from the same patient, as well as on 6 single time-point T-PLL cell samples (5 from indolent-stage T-PLL; Fig. 1A; supplementary Data 1; supplemental Table 1). All samples were treatment-naïve and classified as either indolent or active disease, based on consensus guidelines[2]. Indolent samples were acquired at the time point of diagnosis whereas active T-PLL samples were taken after disease progression and immediately before initiation of treatment (median time between paired samples was 415 days (139–1241), Fig. 1B). Both PB leukocyte counts (median 19.73 G/l (9–29 G/l) vs 91.74 G/l (57.6–369.7 G/l) in indolent vs active T-PLL, $p < 0.0001$, Mann–Whitney–Wilcoxon test (MWW), Fig. 1C) and serum LDH (median 237.5U/l (169-324U/l) vs 693U/l (291-1935U/l), $p < 0.0001$, MWW), markedly differed between both phases. Technical validation of DEGs was performed using qRT-PCR (Pearson's $r = 0.95$, $p = 0.0003$, Supplementary Fig. 1A; supplemental Table 2). Three of the 17 cases (pair_9, single_indolent_2, single_indolent_3) were TCL1A-negative.

After batch correction (Supplementary Fig. 1B) and dimensionality reduction, we observed a high inter-patient heterogeneity among T-PLL cell clusters in contrast to a cell-type specific clustering of non-tumor PBMCs and healthy-donor derived cells (Fig. 1D–E; Supplementary Fig. 1C–D). T-PLL tumor cells were identified based on their expression of T-PLL marker genes (e.g., *CD3D*, *TCL1A*, *TRAC*), and unsupervised cluster allocation (Fig. 1F). Cell labels of non-T-PLL cells were independently predicted via projection onto a reference data set of healthy-donor derived PBMCs[34] and presented cell type-specific clustering in the t-SNE representation (Fig. 1G). Investigating the relative frequencies of cell types, we observed a high relative tumor load within all samples (median purity 92.4% of T-PLL cells, Fig. 1H), except for patient pair_6 (indolent: 55.6%; active: 34.5%) and the indolent samples of pair_9 (27.6%) and pair_10 (32.5%). The predicted cell types were confirmed by their expression of well-established lineage markers (Fig. 1I).

Pseudo-bulk differential gene expression analysis, conducted between active time-point derived T-PLL cells ($n = 12$ samples) and healthy-donor T cells ($n = 10$ samples, supplementary Figs. 2A–C), revealed a strong correlation of DEGs when compared to bulk gene expression data sets[13,35] (rho=0.7074–0.7592, Spearman).

### Progression-associated transcriptomic shifts are heterogeneous at the individual gene level, but display shared patterns of pathway deregulation

Pairwise differential gene expression analysis (Fig. 2A) retained a total of 1019 genes that were significantly deregulated in the T-PLL cells of at least one sample pair (absolute log2FC ≥ 0.5, FDR < 0.05). The overall number of DEGs varied strongly among pairs (range: 1-458 DEGs). Importantly, most (61.92%) of the identified DEGs were patient-specific, although there was a large set of DEGs that appeared in at least 2 sample pairs (38.08%; $p < 0.0001$, permutation test; Supplementary Fig. 3A–B). These included major regulators and components of cell metabolism (*TXNIP*, *COX6C*), of TNFα/NFκB signaling (*TNFAIP3*, *NFKBIA*), and of cellular senescence (*GADD45B*, *ZFP36L2*). Notably, DEG clustering identified two groups of patients revealing opposite patterns of expression changes within some of the most recurrently deregulated genes (Fig. 2B). The strongest differences were observed in TNFα/NFκB

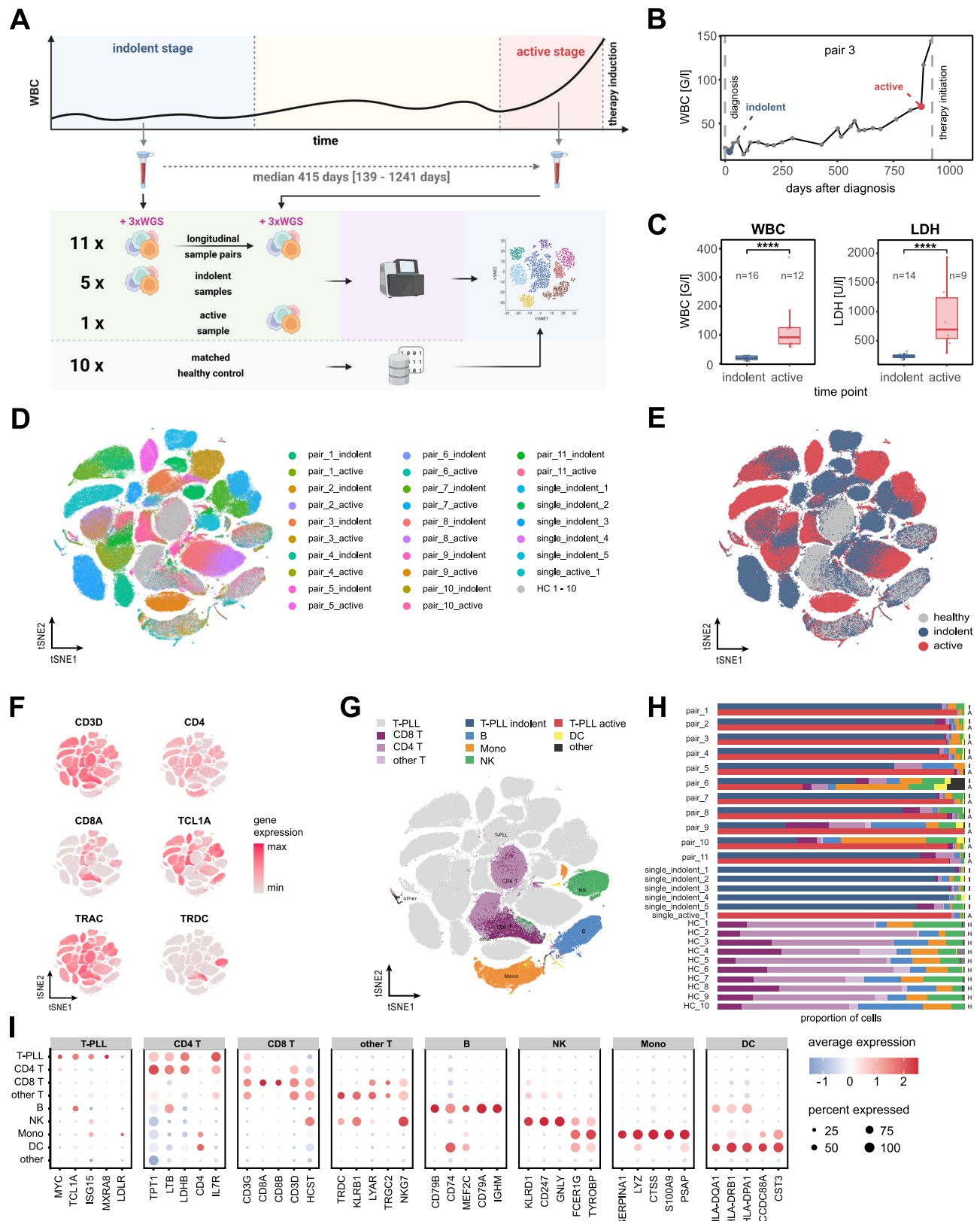

signaling-associated genes, with a severe downregulation in the active time-point derived sample in those patients that were characterized by a comparably strong NFκB-signature during the indolent T-PLL stage (Supplementary Fig. 4A–F). Overall, the most uniform shifts of gene expression across all indolent/active sample pairs were detected in GSEA-confirmed functional gene sets[36,37] associated with cell metabolism and cell-cycle regulation (Supplementary Fig. 5A–C).

Based on the observation of conserved T-PLL cell subclusters within paired samples (Supplementary Fig. 5D), we next investigated differential gene expression between T-PLL cell clusters that expanded and those that retracted over time (Fig. 2C). Interestingly, in contrast to the heterogeneous deregulation of single genes, GSEA indicated a higher degree of homogeneity in deregulated pathways. Most prominent was a strong enrichment of gene sets related to cell cycle (e.g.,

**Fig. 1 | Single-cell transcriptome atlas of T-PLL peripheral blood mononuclear cells. A** Experimental set-up of single-cell RNA sequencing in T-PLL: longitudinal peripheral blood sample pairs from 11 T-PLL patients were collected during indolent disease stage and after transition to active T-PLL. Additional unpaired samples were obtained from 5 indolent T-PLL patients and 1 active T-PLL[2]. PBMCs were isolated and submitted to 3′ single-cell RNA sequencing. Publicly available data of 10 age- and sex-matched healthy donors[73] were integrated to create the final scRNA data set. PBMCs of 3 longitudinal sample pairs were additionally subjected to whole genome sequencing (WGS). Created in BioRender. Herling, M. (2026) https://BioRender.com/1mp0lno **B** Exemplary tumor development of a primarily indolent T-PLL patient. White blood cell count (WBC) is illustrated over time. Samples were acquired at day 20 (indolent stage, blue) and day 874 (active stage, red) after diagnosis. **C** Comparison of established hematologic disease markers between indolent and active-stage T-PLL. WBC (mean 20.11 vs 120.57 G/l, indolent vs active T-PLL, $p < 0.0001$, Mann–Whitney–Wilcoxon test (MWW), two-sided) and serum lactate dehydrogenase (mean 236.7 vs 876.9 U/l, $p < 0.0001$, MWW, two-sided) were significantly increased in active T-PLL samples.

Box plots: center: 50th percentile, box bounds: 25th and 75th percentiles (IQR), whiskers: smallest and largest observations within 1.5×IQR of the box. **D** t-SNE representation of batch-corrected PBMC single-cell gene expression data. Cell colors indicate the sample of origin. **E** t-SNE representation of PBMC data sets color-coded by type of sample (gray: healthy donor, blue: indolent T-PLL, red: active T-PLL). **F** Single-cell expression of T-PLL marker genes in the PBMC data set[19]. Colors indicate relative gene expression (light gray: minimal observed expression, red: maximum observed expression). **G** t-SNE projection of unsupervised-predicted cell type labels in the PBMC data set. Colors represent the assigned major cell types. **H** Relative contribution of different cell types (color-coded, see (**G**)) among samples. Letters on the right $y$-axis indicate the type of sample (H: healthy donor, I: indolent T-PLL sample, A: active T-PLL sample). **I** Dot plot visualization of hematologic marker gene expression among different cell types. Dot colors indicate scaled normalized expression values (blue: low relative expression, red: high relative expression). Dot sizes represent the percentage of cells within each cell type expressing the respective gene. Source data and complete summaries of statistical analyses are provided in the Source Data file.

HALLMARK E2F targets or G2M checkpoint, 8/11 pairs), energy metabolism (e.g., HALLMARK oxidative phosphorylation, 7/11 pairs), and the transcription factor MYC (e.g., HALLMARK MYC targets V1 or V2, 8/11 pairs) in the expanding T-PLL cell clusters (Fig. 2D; Supplementary Fig. 5E–G). Pairwise pseudotime analyses further underlined these patterns (Supplementary Fig. 6A–B).

### Clonal expansion of ATM lesions and MYC CN gains in active-stage T-PLL

To identify genomic lesions potentially driving the transcriptional changes observed in active T-PLL, we performed WGS on three longitudinal sample pairs. While the overall mutational burden remained largely unchanged (mean number of short variants: 11.83/MB vs 11.62/MB, indolent vs active-stage T-PLL, $p = 1$, MWW, Fig. 2E; Supplementary Fig. 7A, supplementary Data 2), we detected an enrichment of several variants from indolent to active stage, especially in genes reported to be recurrently mutated in T-PLL[13] (Supplementary Fig. 7B, Fig. 2F). Most notably, we observed a marked increase in the allele frequency of mono-allelic *ATM* lesions in all three patients (mean AF: 69.76% vs 91.33%, indolent vs active-stage T-PLL, Fig. 2G). In contrast, subclonal mutations of *JAK/STAT* pathway genes, which were detected at all indolent time points, did not show consistent outgrowth in active T-PLL (Supplementary Fig. 7C).

CN inferences based on gene expression highlighted T-PLL-typical aberrations, i.e., chromosome 8q gains, 11q deletions, and alterations of chromosomes 14 and 17 (Supplementary Fig. 8A). These hallmarks, including the 8q gains, were already present in the indolent time-point derived samples, suggesting an early role in disease evolution. The CN abundance of other chromosomal regions varied between indolent and active disease samples due to the outgrowth of different T-PLL subclones. Across all samples, there was no significant difference in the proportion of the genome affected by CN alterations between indolent and active T-PLL (Supplementary Fig. 8B–D). These findings were corroborated by WGS-based structural variant analysis, which confirmed that several key genomic lesions, such as juxtaposition of TCL1A and TCRAD, were already present at the indolent disease stage (Supplementary Fig. 9A–B). The most prominent genomic alteration associated with progression to the active stage was a gain of chromosome 8q, encompassing the MYC locus at 8q24.21 (Fig. 2H). Notably, CN variants (CNVs) identified by WGS showed a strong concordance with those inferred from gene expression profiles (Supplementary Fig. 9C), supporting the reliability of the transcriptome-based CNV calls.

### Amplified MYC-signatures as a key feature in the transition from indolent to active-stage T-PLL

Given the identification of prominent MYC gene sets in expanding clones and the observed copy number gains of *MYC* in active-stage

T-PLL, we further interrogated the relevance of MYC-regulated genes in the transition from indolent to active disease. We derived cell-wise expression scores based on the HALLMARK gene sets MYC targets V1 and V2[36]. Sample-based analysis indicated a marked difference in their expression not only between healthy-donor derived T cells[33] and active T-PLL, but especially in the transition from indolent to active disease (Fig. 3A; Supplementary Fig. 10A). Importantly, the upregulation of MYC targets was further increased in expanding T-PLL subclusters in 8/11 sample pairs (72.7%, Fig. 3B; Supplementary Fig. 10B). In line with the enhanced expression of MYC target genes was an overall upregulation of *MYC* mRNA from healthy-donor T cells over indolent to active T-PLL ($p < 0.0001$, MWW, Fig. 3C; Supplementary Fig. 10C). Significant upregulation of *MYC* mRNA in T-PLL was validated in independent bulk-analyzed data sets of 117 T-PLL ($n = 69$ by expression arrays[13], $n = 48$ by RNAseq[35]; Fig. 3D).

CN inferences revealed increases in the mean number of *MYC* alleles within several sample pairs (7/11; Fig. 3E), especially within expanding subclusters, which also correlated with *MYC* mRNA expression (rho=0.268, $p < 0.0001$, Spearman; Supplementary Fig. 10D–E). Importantly, the increase of *MYC* mRNA levels translated into upregulation of MYC target genes ($r = 0.945$, $p < 0.0001$, Pearson; Fig. 3F; Supplementary Fig. 10F). Elevated *MYC* levels in T-PLL (over age-matched healthy-donor CD3 + T cells) were verified at the protein level ($p = 0.014$, MWW). Increases of MYC protein in the intra-individual indolent-to-active comparison were observed in 3/3 analyzed sample pairs (Fig. 3G–H; Supplemental Table 3). Observed protein expression was also positively associated with mRNA expression of MYC target genes (Fig. 3I).

Cluster-based correlation of gene expression scores provided implications of these prominently upregulated MYC expression signatures, i.e., by revealing strong positive associations between MYC-target expression scores and processes of energy metabolism, such as oxidative phosphorylation and glycolysis (for both rho≥0.67, $p < 0.0001$, Spearman), within both shrinking and expanding T-PLL subclusters (Fig. 3J; Supplementary Fig. 10G).

### Upregulated energy metabolism and ATP generation characterize the progression to active T-PLL

Having detected recurrent alterations in metabolism-associated genes in conjunction with upregulated MYC (and its networks, e.g., *TXNIP*, *ZFP36*, *UQCR10*, *FTH1*) and given the central role of MYC as a metabolic regulator in cancer[38–42], we examined energy metabolism within T-PLL cells in more detail. Both oxidative phosphorylation and glycolysis gene set expression scores were upregulated in active time-point derived T-PLL (Fig. 4A). Remarkably, T-PLL cells from the indolent time point presented mean expression scores of oxidative

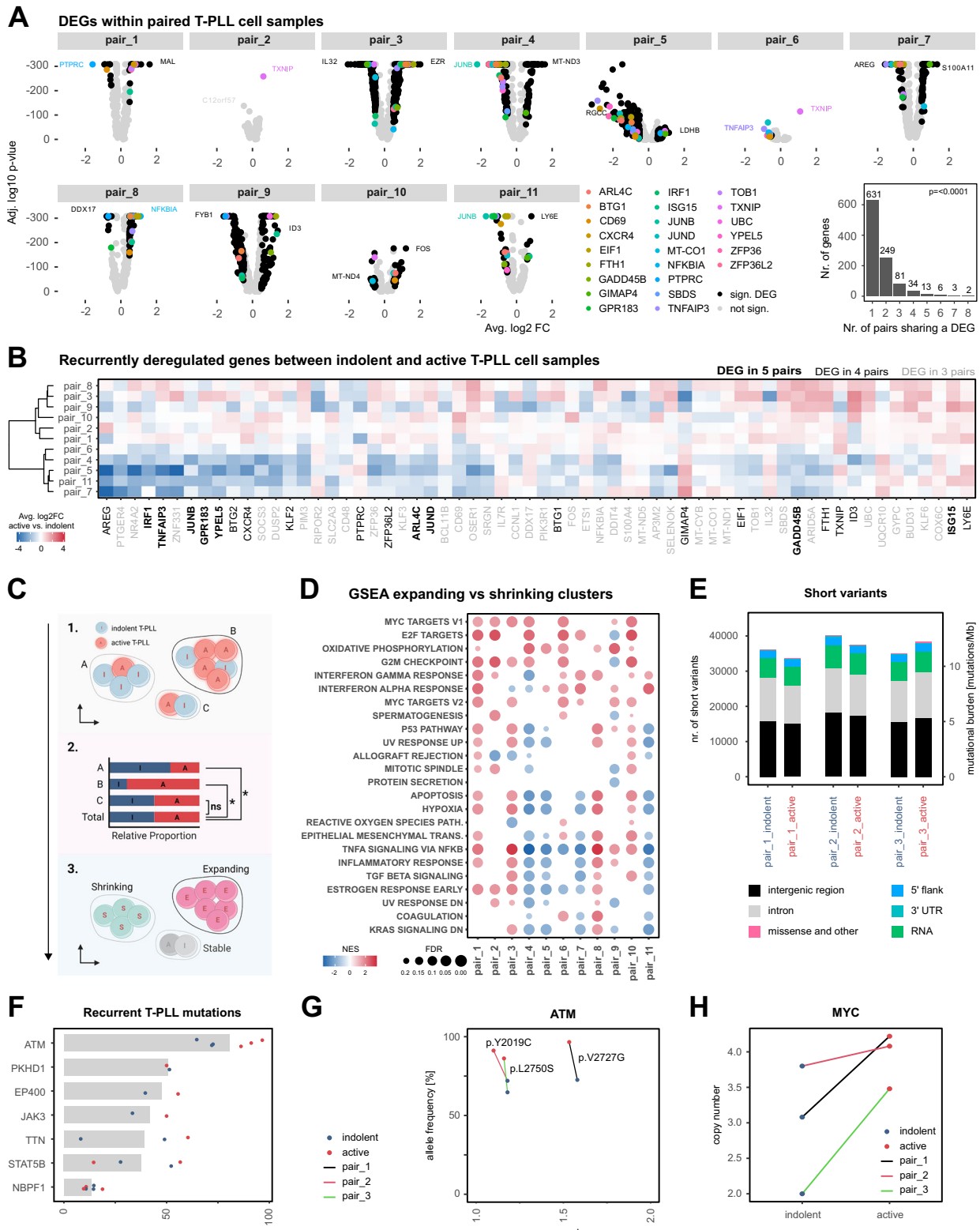

**A** DEGs within paired T-PLL cell samples

**B** Recurrently deregulated genes between indolent and active T-PLL cell samples

**C**

**D** GSEA expanding vs shrinking clusters

**E** Short variants

**F** Recurrent T-PLL mutations

**G** ATM

**H** MYC

phosphorylation similar to those of healthy-donor derived T cells, suggesting that the observed strong upregulation of this gene set in active-stage T-PLL cells is a specific feature of the transition from indolent to active T-PLL. These results were confirmed by cluster-based analyses, in which expanding clusters displayed a significant increase in oxidative phosphorylation or glycolysis scores as compared to shrinking T-PLL cell clusters in 7/11 pairs (Supplementary Fig. 11A–B).

There was also a robust interaction between both energy metabolism pathways (rho=0.398, *p* < 0.0001, Spearman; Fig. 4B), indicating a general metabolic activation and increased ATP synthesis in T-PLL cells during disease progression. Investigating the relative expression of transcripts associated with cell metabolism revealed an increase from indolent to active T-PLL (Fig. 4C; Supplementary Fig. 11C). A significant general upregulation of transcripts encoding for metabolic genes in T-PLL vs healthy-donor derived T cells was further confirmed

**Fig. 2 | Active T-PLL samples present a significant enrichment of pathways involved in MYC signaling, cell cycle regulation, and energy metabolism.**
**A** Volcano plots: pairwise differential gene expression analyses between T-PLL cells from active vs indolent stages. Genes with an absolute log2 expression fold-change ≥0.5 and adjusted $p < 0.05$ were considered significantly different. Colored points indicate differentially expressed genes (DEGs, $n = 24$ genes) detected in at least 5/11 sample pairs. Positive fold-changes indicate elevated expression in active-disease T-PLL. sign.: significant. Bar chart: number of recurring DEGs. A significant proportion of DEGs were detected across multiple sample pairs ($p < 0.0001$, permutation test, B = 10,000, one-sided). **B** Differential expression heatmap of recurrent DEGs. Color indicates the log2 fold-change between active and indolent T-PLL cells (red: upregulated in active T-PLL, white: no change, blue: downregulated in active T-PLL). Font of gene labels indicates the number of patients that showed deregulation in the same direction (bold black: DEG in 5 pairs, regular black: DEG in 4 pairs, regular gray: DEG in 3 pairs). **C** Experimental design for the identification of shrinking and expanding clusters. For each longitudinal sample pair, SNN-based clustering on integrated T-PLL cell gene expression was performed[34]. The relative

contributions of each time point to the individual clusters were quantified and compared to the global indolent/active cell ratio using permutation testing (B = 10,000, two-sided). Clusters that significantly differed from the overall sample distribution (false discovery rate (FDR) < 0.05, observed log2 difference in proportion ≥1) were identified as expanding or shrinking. Remaining clusters were labeled as stable. Created in BioRender. Herling, M. (2026) https://BioRender.com/eih43ig **D** GSEA enrichment of most abundant HALLMARK pathways between expanding and shrinking clusters[36]. Dot colors represent the Normalized Enrichment Score (NES, red: enrichment in expanding clusters, blue: enrichment in shrinking clusters). Dot sizes indicate the FDR. **E** Total number and predicted type of short variants derived from WGS. **F** Mean variant allele frequencies (VAFs) of functional coding variants in genes previously identified as recurrently mutated in T-PLL ($n = 6$ T-PLL samples)[13]. **G** Dot plot comparing WGS-derived VAFs of *ATM* mutations and copy number alterations (CNAs) between three longitudinal T-PLL sample pairs. **H** WGS-derived CNAs of *MYC* in three sequential T-PLL patients. Source data and complete summaries of statistical analyses are provided in the Source Data file.

in two independent bulk gene expression data sets of 117 T-PLL[13,35] (Fig. 4D).

To assess the functional relevance of the observed transcriptomic metabolic activation, we dynamically recorded energy metabolism in our samples using the Seahorse XFe 96 extracellular flux analyzer. Importantly, indolent T-PLL demonstrated low basal activity of glycolysis and respiration as well as a strongly restricted capacity to upregulate their energy metabolism upon TCR stimulation, when compared to age-matched healthy-donor derived T cells (Fig. 4E–F). In contrast, T-PLL cells of the active time point presented a significant increase of basal bioenergetics and surpassed the metabolic confinement observed in indolent T-PLL cells. In line with these functional data, both ex-vivo respiratory capacity and glycolytic reserve were positively correlated in our samples to the gene expression scores of the HALLMARKs oxidative phosphorylation and glycolysis, respectively (Supplementary Fig. 11D–E).

## Active-disease T-PLL cells show attenuated signatures by exogenous growth signals, e.g., TCR stimuli

Our indolent/active comparative setup also allowed to assign central growth-regulating pathways, particularly those affected by TCL1A, i.e., TCR signaling[15,19], a stage-specific relevance. Therefore, we investigated the gene expression of TCR-signaling components and their down-stream signatures of transcriptomic responses in our data set. Using the DoRothEA regulon collection[43] to infer transcription factor activities, we found that transcription factors downstream of TCR signaling were among the most profoundly deregulated when comparing indolent with active T-PLL (Supplementary Fig. 14A). Notably, a strong activation of TCR-signaling signatures was particularly evident within indolent-time point T-PLL cells (Fig. 5A), predominantly observed for the transcription factors ATF2, REL, JUN, and RELB.

Consistent with these findings, we confirmed that aberrations of proximal negative regulators of TCR-signaling input that are known to be downregulated in T-PLL, namely *CTLA4* and *LAG3*[13,19], were already present in all indolent samples (Fig. 5B), strongly suggesting that these changes might contribute to early T-PLL pathogenesis.

Moreover, highlighting a diminished relevance of TCR activation in the active disease stage, we observed downregulations of TCR-signaling components and of downstream transcription factors during disease progression in multiple sample pairs. The expanding clusters of patient pair_3 showed a uniform downregulation of the TCR complex including all CD3 subunits, *TRBC2*, and the phosphatase *PTPRC* (CD45; all $p < 0.0001$; Fig. 5C–D). A strong decrease in *IL32* expression (log2FC = −2.062, $p < 0.0001$) further pointed towards an overall decrease in cellular immune activation in this patient.

Tumor cells of patient pair_1 displayed similar aberrations during their transformation to active-stage T-PLL. Here, we observed a

marked expansion of a subcluster harboring a complete down-regulation of *PTPRC* (CD45, Fig. 5E). Correlating with this aberration, T-PLL cells acquired marked deregulations of other genes well-associated with cancer, including *MAL*, *LY6E*, and *COX6C*. Functionally, this corresponded to an overall decreased ex-vivo TCR response that specifically affected the CD45-negative subcluster. In fact, CD3/CD28 crosslinking resulted in lower induction of T-cell activation markers on CD45-negative cells and in a reduced phosphorylation of TCR-downstream kinases in active-stage T-PLL (Fig. 5F–G, Supplementary Figs. 12–13). In our cohort, downregulation or loss of PTPRC was observed in 4/11 (36.4%) T-PLL. Overall, these results demonstrate diminished TCR signaling in the indolent-to-active-stage transition of T-PLL.

## Underrepresented immune-response pathways and reduction of cell interactions with the tumor-microenvironment implicate further mechanisms of autonomy

We next aimed to characterize the non-neoplastic cells in T-PLL. In the 28 samples analyzed here, the transcriptome profiles set apart distinct subsets of PB cell types (Fig. 6A). Investigating the kinetics in the compositions of non-T-PLL cells over disease progression, we observed most distinct changes in the proportion of monocytes ($p = 0.056$, MWW, Fig. 6B–C) and dendritic cells ($p = 0.076$, MWW, Supplementary Fig. 14B) with their relative increase in PBMCs of active T-PLL. Analyses of PB monocyte counts derived from hematology laboratory parameters available for 31 patients confirmed an elevated absolute number of monocytes in the 22 active T-PLL patients ($p = 0.0005$, MWW, Supplementary Fig. 14C). Further characterization revealed a significant increase in the CD14/CD16 ratio from indolent to active T-PLL patient samples ($p < 0.0001$, chi-squared test; Supplementary Fig. 14D).

To identify transcriptional programs associated with disease progression in the tumor microenvironment, we performed pairwise differential expression analysis between indolent and active-stage PBMC subsets. This revealed recurrent patterns of deregulated gene expression (Fig. 6D). Deregulations of the AP-1 subunits (*JUN*, *JUNB*, *JUND*) were prominent in PBMCs of the active time point. Similar to T-PLL tumor cells, we also detected deregulated TNFα/NFκB-signaling components and interferon-α response associated genes (*IFI6*, *IFI44L*, *MNDA*, *NFKBIA*) in PB monocytes of active T-PLL samples. Similar results were observed in a more conservative pseudo-bulk approach including all 28 T-PLL samples (Supplementary Fig. 15A).

GSEA revealed significant global alterations at the level of functional pathway clusters, including downregulation of inflammation-associated gene sets among nearly all cell types, alongside recurrent upregulation of interferon-alpha signaling and cell-cycle genes (Fig. 6E). In agreement with these dynamics, we observed reduced cell-

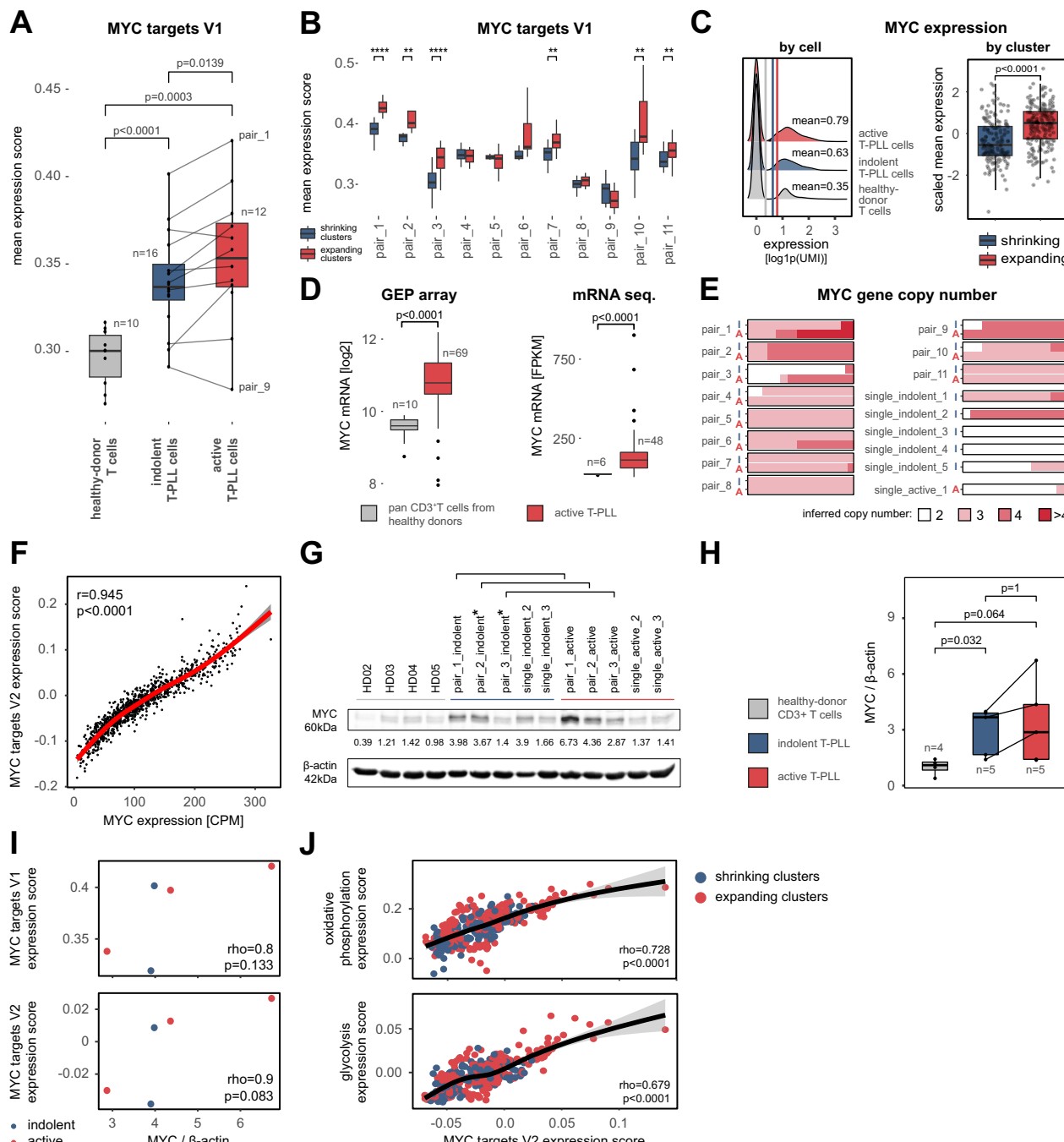

**Fig. 3 | Enhanced MYC target expression is associated with T-PLL progression.**
**A** Mean expression scores of the HALLMARK MYC targets V1 gene set[34]. Upregulation from indolent to active T-PLL was observed in 9/11 patients ($p = 0.0139$, permutation test, B = 10,000, two-sided). **B** Mean HALLMARK MYC targets V1 expression scores of shrinking and expanding T-PLL cell clusters (****$p < 0.0001$, ***$p < 0.001$, **$p < 0.01$, *$p < 0.05$, MWW, two-sided). **C** Increase of *MYC* mRNA expression during T-PLL development. Left: Cell-based expression levels of *MYC* mRNA in healthy-donor derived T cells[73], indolent-stage, and active-stage T-PLL cells. Right: Scaled mean *MYC* mRNA expression in shrinking ($n = 172$) vs expanding ($n = 216$) T-PLL cell clusters ($p < 0.0001$, MWW, two-sided). **D** Significant upregulation of *MYC* in two independent T-PLL data sets (each $p < 0.0001$, MWW, two-sided, left: gene expression array data of 69 T-PLL compared to CD3 + T cells of 10 healthy donors[13], right: mRNA sequencing of 48 T-PLL and CD3 + T cells of 6 healthy donors[35]). **E** Relative distribution of inferred *MYC* copy number states in T-PLL cells. 7/11 T-PLL pairs (63.6%) presented an increase of *MYC* copy number-elevated T-PLL cells from indolent to active disease. (I: indolent, A: active) **F** Significant correlation of *MYC* mRNA expression and HALLMARK MYC targets V2 expression scores in $n = 1000$ T-PLL cell pseudo-bulks ($p < 0.0001$, $r = 0.945$, Pearson

correlation, two-sided, gray band: 95% confidence interval). **G** Western blot of MYC protein expression in age-matched healthy-donor derived T cells ($n = 4$, gray line), indolent ($n = 5$, blue line), and active T-PLL ($n = 5$, red line). **H** Densitometrically quantified MYC protein expression from (**G**). Upregulated MYC protein expression in indolent ($p = 0.032$, MWW, two-sided) and active-stage T-PLL cells ($p = 0.064$, MWW, two-sided) compared to healthy-donor derived T cells. *PBMCs were collected not from the original sample at diagnosis, but shortly thereafter.
**I** Correlation of MYC protein levels to mean HALLMARK MYC target gene expression scores (MYC targets V1: rho=0.8, $p = 0.133$; MYC targets V2: rho=0.9, $p = 0.083$, Spearman correlation, two-sided). **J** Mean HALLMARK MYC targets V2 gene expression scores correlated to mean expression scores of energy metabolism gene sets in shrinking and expanding T-PLL clusters (HALLMARK oxidative phosphorylation: rho=0.728, $p < 0.0001$; HALLMARK glycolysis: rho=0.697, $p < 0.0001$, Spearman correlation, two-sided). Gray band: 95% confidence interval. Definition of box plots: center: 50th percentile, box bounds: 25th and 75th percentiles (IQR), whiskers: smallest and largest observations (**A**) within 1.5×IQR of the box (**B**–**D**, **H**). Source data and complete summaries of statistical analyses are provided in the Source Data file.

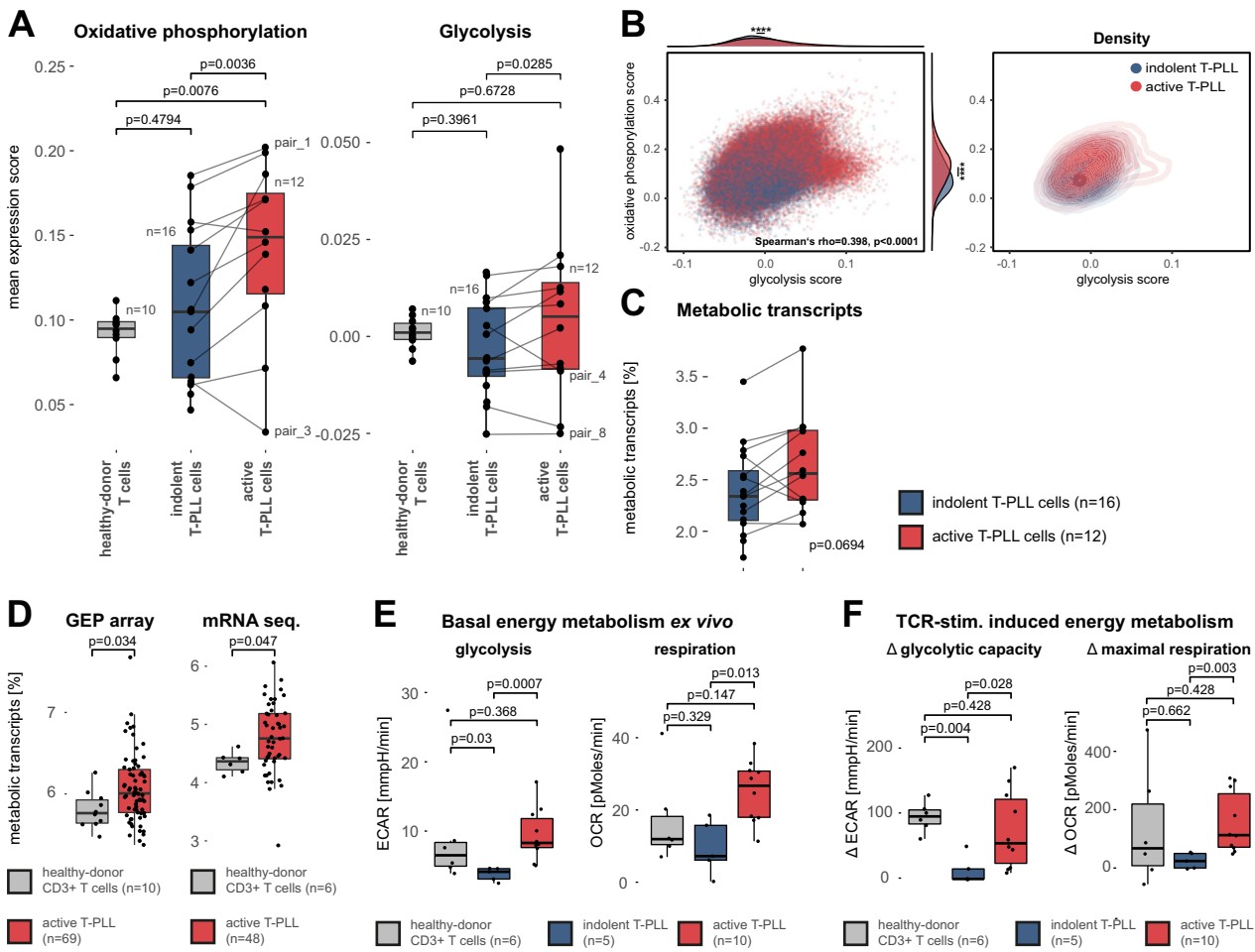

**Fig. 4 | Upregulated energy metabolism as a common feature in T-PLL progression. A** Mean expression scores of energy metabolism gene sets (left: HALLMARK oxidative phosphorylation, right: HALLMARK glycolysis)[36,73]. Oxidative phosphorylation and glycolysis-associated genes are strongly upregulated in active vs indolent T-PLL cells (oxidative phosphorylation: $p = 0.0036$; glycolysis: $p = 0.0285$, permutation test, B = 10,000, two-sided). **B** Correlation of HALLMARK oxidative phosphorylation and HALLMARK glycolysis gene expression scores within T-PLL cells. Left panel: Per-cell representation of expression score values (blue: indolent T-PLL cells, red: active T-PLL cells). Density distributions of T-PLL cells were compared for each score using two-sided MWW tests (****$p < 0.0001$). Both expression scores were positively correlated (rho=0.398, $p < 0.0001$, Spearman, two-sided). Right: Contour chart illustrating the two-dimensional density distribution of cell expression scores between indolent (blue) and active (red) T-PLL cells. **C** Mean percentage of transcripts encoding metabolism-associated genes in all protein-coding transcripts. Active-stage T-PLL cells show a relative increase of metabolic transcripts compared to indolent stage T-PLL cells ($p = 0.0694$, permutation test, B = 10,000, two-sided) **D** Relative increase of metabolic transcripts in publicly available T-PLL bulk gene expression data sets. Left: Gene expression array data comparing CD3 + T cells from 10 healthy donors to

69 T-PLL samples[13] (median proportion of metabolic transcripts: 5.802 vs 6.073%, $p = 0.034$, MWW, two-sided). Right: mRNA sequencing data set of CD3[+] T cells from 6 healthy donors and 48 T-PLL samples[35] (median proportion of metabolic transcripts: 4.365% vs 4.758%, $p = 0.047$, MWW, two-sided). **E** Basal energy metabolism in indolent and active-stage T-PLL samples compared to age-matched healthy-donor derived pan CD3 + T cells. Agilent Seahorse assays were used to analyze energy metabolism ex-vivo (statistics: unpaired MWW, two-sided). Left panel: Box-whisker plot of basal glycolysis. Right: Box-whisker plot of basal respiration. **F** Maximal ex-vivo energy metabolism of T-PLL cells vs age-matched healthy-donor derived CD3 + T cells upon TCR stimulation via CD3/CD28 crosslinking (normalized to basal levels). Left panel: Box-whisker plot of maximum glycolytic capacity. Right: Box-whisker plot of maximal respiration. Statistical comparison using unpaired two-sided MWW. Indolent-phase T-PLL cells were strongly restricted in their capability to upregulate energy metabolism upon TCR stimulation. Definition of box plots: center: 50th percentile, box bounds: 25th and 75th percentiles (IQR), whiskers: smallest and largest observations (**A**) within 1.5×IQR of the box (**C**–**F**). Source data and complete summaries of statistical analyses are provided in the Source Data file.

cell interactions between T-PLL cells and PBMC cell types in active-stage T-PLL (Fig. 6F; Supplementary Fig. 15B). Contrasting this overall pattern, signaling from T-PLL cells to monocytes was markedly enhanced in most active samples.

To explore key ligands involved in the remodeling of the tumor microenvironment, we performed NicheNet analysis in longitudinal sample pairs (Fig. 6H). Among the most recurrent ligands identified was Annexin A1 (*ANXA1*), a central player in anti-inflammatory signaling[44,45]. In our data set, ANXA1 was of particular importance in the T-PLL-monocyte interaction (Supplementary Fig. 15C–D). In addition, recurrent downregulation of *CD48* in active-stage T-PLL cells (9/11

T-PLL pairs) emerged as an important contributor to the transcriptomic changes in the tumor microenvironment.

On a broader scale, we noted a distinct downregulation of antigen presentation pathways across non-malignant immune subsets, particularly affecting multiple HLA class II genes (Fig. 6I). Analysis of genes associated with the widely observed upregulation of interferon-α signaling in immune cells of active-stage T-PLL samples identified prominent upregulation of *IFI44* and *IFI44L*, as well as *LY6E* (Fig. 6J).

Together, these findings suggest a progressive dampening of anti-tumor immune responses from indolent to active

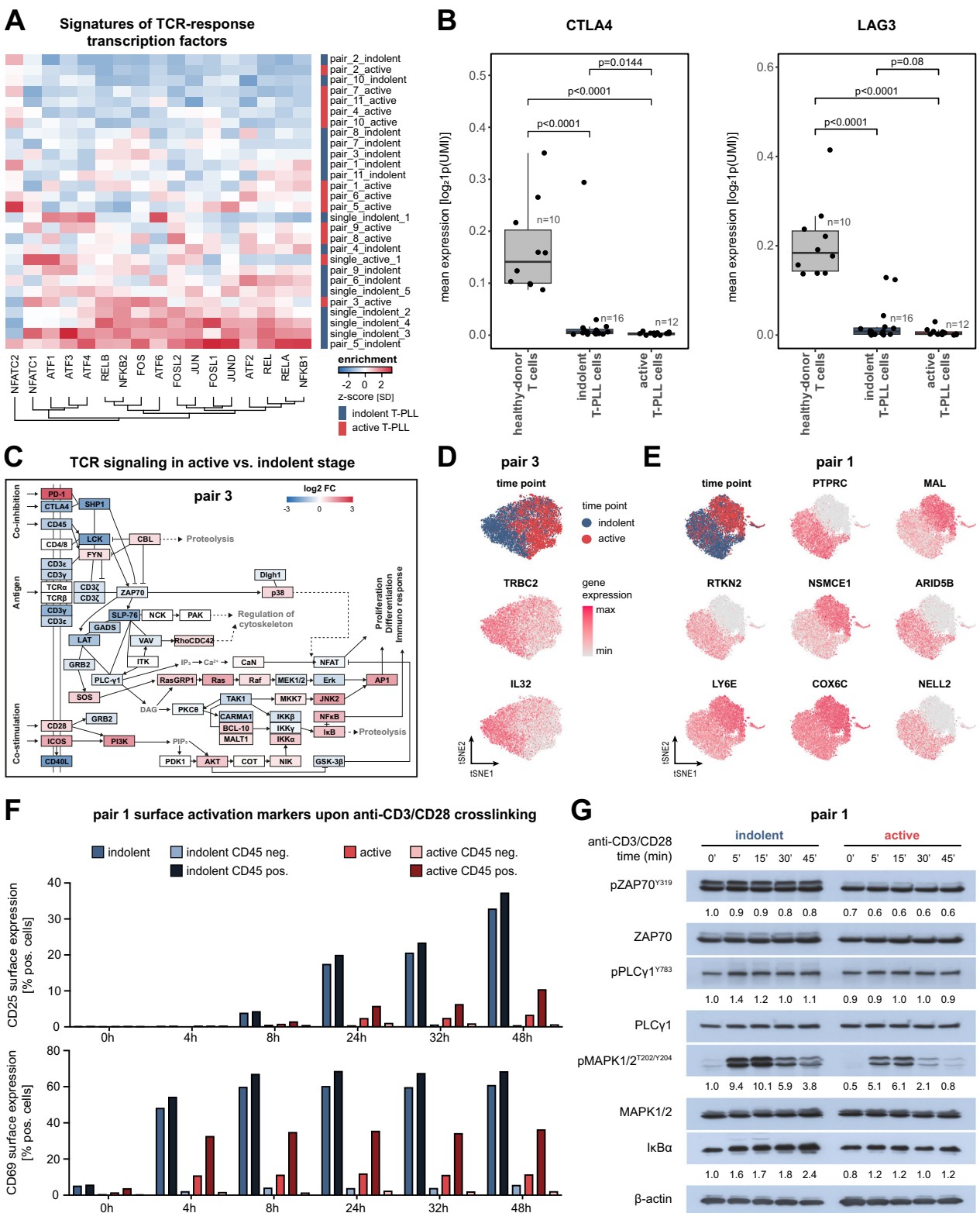

**A** Signatures of TCR-response transcription factors

**B** CTLA4 LAG3

**C** TCR signaling in active vs. indolent stage

**D** pair 3

**E** pair 1

**F** pair 1 surface activation markers upon anti-CD3/CD28 crosslinking

**G** pair 1

T-PLL, marked by reduced antigen presentation and globally diminished cell-cell interactions. These processes were significantly associated with deregulations of TNFα/NFκB and interferon-α signaling in both, the leukemic and non-neoplastic cell compartments.

## Discussion

In a cohort of longitudinally acquired T-PLL samples per patient at their indolent and active leukemic phase, we leveraged on the strengths of single-cell RNA sequencing to illustrate the dynamic transcriptomic and cell-compositional changes in the context of tumor progression. Our data contribute to a model of T-PLL evolution. Therein, we established a strong upregulation of MYC gene expression signatures, metabolic activation, progressive downregulation of immune-response pathways, and diminished interaction with non-neoplastic cells as critical features of active-stage T-PLL. Ultimately, these processes might promote increased cellular autonomy towards leukemic expansion (Fig. 7).

**Fig. 5 | Declining role of TCR-mediated signaling in active-stage T-PLL cells.**
**A** Heatmap of scaled TCR-pathway transcription factor signatures in indolent and active-stage T-PLL cells as derived from the DoRothEA gene regulatory network[43]. Samples were arranged by their mean enrichment across all TCR transcription factor signatures. **B** Mean gene expression of negative TCR-signal regulating checkpoint molecules *CTLA4* and *LAG3* shows significant downregulation in indolent and active-stage T-PLL cells as compared to healthy-donor derived CD3 + T cells[73] (permutation test, B = 10,000, two-sided). Box plots: center: 50th percentile, box bounds: 25th and 75th percentiles (IQR), whiskers: smallest and largest observations within 1.5×IQR of the box. **C–D** Downregulation of TCR-signaling molecules in active T-PLL cells of patient pair 3. **C** Differential gene expression of TCR-signaling cascade components. Genes and gene relationships were derived from KEGG[37]. Gene color represents the log2 expression fold-change between active and indolent-stage T-PLL cells (red: upregulation in active-stage T-PLL, white: no difference, blue: downregulation in active-stage T-PLL). *P*-values from two-sided MWW with correction for multiple testing. **D** Single-cell gene expression values of TCR-signaling associated genes in patient pair 3. Cell stage (indolent: blue, active: red) and scaled gene expression values (red: maximum

expression, light gray: minimum expression) are color-coded on the t-SNE representation of pair 3. **E** Single-cell gene expression values of highly deregulated oncogenes and tumor-suppressor genes in patient pair 1: T-PLL stage (indolent: blue, active: red) and scaled gene expression values (red: maximum expression, light gray: minimum expression) are color-coded on the t-SNE representation of pair 1 T-PLL cells. **F** Surface expression of activation markers upon TCR stimulation via anti-CD3/CD28 crosslinking in CD45+ and CD45- T-PLL cells of patient pair 1 assessed by flow cytometry. Cells were stratified by T-PLL disease stage and by their expression of CD45. CD45- T-PLL cells showed diminished expression of surface activation markers upon TCR activation. CD45 + T-PLL cells of the active T-PLL stage displayed a reduced response to TCR-signaling compared to indolent CD45 + T-PLL cells. See Supplementary Fig. 12 and Supplementary Fig. 13 for gating strategies. **G** Representative Western blot (*n* = 1) showing reduced phosphorylation of TCR-downstream kinases in active-stage T-PLL samples of patient pair 1 upon CD3/CD28 crosslinking. Densitometric quantification via normalization to β-actin signals of unstimulated indolent T-PLL sample. Source data and complete summaries of statistical analyses are provided in the Source Data file.

---

CN gains and epigenetic mechanisms mediate the upregulation of the transcription factor and oncogene MYC in >80% of T-PLL patients[13,35,46]. Overexpression of *MYC* is also particularly prominent at the exponential leukemic phase in the *Lck^{pr}-hTCL1A^{tg}* T-PLL mouse model[13]. In healthy T cells, MYC exerts indispensable functions in cell-cycle regulation and metabolism[47]. MYC constitutes a primary mediator of TCR signaling-induced metabolic activation, leading to a reprogramming towards high glucose and glutamine metabolism[39,48,49]. In light of these data, the implications of our findings in almost all paired indolent/active samples (even in those with few DEGs), namely of strongly increased MYC signatures paralleled by upregulation of energy metabolism, suggest a central role of MYC in the principles underlying T-PLL progression. Further experiments have to establish if MYC is a primary causative driver or effector of these changes, and to dissect the specific molecular links.

The inferred survival advantage by upregulated MYC signatures was well-illustrated here by the progressive kinetics of MYC-RNA-high or MYC-CN-high T-PLL subclusters, as well as by further increases of MYC protein levels from indolent to active disease stage. Although our data suggest *MYC* amplification and elevated activity to be an early event in T-PLL pathogenesis, we observed a progressive enhancement, especially within its affected downstream signatures and its oncogenic function during disease activation. This renders MYC networks attractive interventional targets in active T-PLL. Although direct inhibition of MYC has long been considered infeasible[50], this view is shifting with the development of new MYC inhibitors, including a promising compound recently evaluated in a phase-I clinical trial[51]. To date, these inhibitors have not been tested in T-PLL. However, a MYC-dependent efficacy of CDK-inhibitors alongside a therapy-induced downregulation of MYC and its target genes in T-PLL cells have been demonstrated[52,53]. Consequently, besides MYC itself, also its regulators and effectors[54,55] will have to be interrogated more systematically as targets to overcome the treatment resistance inherent to T-PLL.

Our comparative analyses also strongly implicate that signals through the MHC/antigen-TCR axis, the main determinant of growth, differentiation, and inter-clonal homeostatic control in T cells, lose importance during T-PLL progression. This is substantiated by downregulation of TCR-cascade components, by mitigated TCR-stimulation induced responses, and by a reduced activity of TCR-signaling induced transcription factors, all in expanding and active-stage T-PLL cells. Generally, TCR-mediated survival signals are considered crucial in the early stages of T-PLL development and the causal involvement of the kinase co-activator TCL1A, of other T-PLL oncogenes, or of micro-RNAs in enhanced TCR-signaling is well established[15,19,35,56]. This is further supplemented by our finding that negative regulators of TCR activation, namely *CTLA4* and *LAG3*, are already impaired at the indolent

disease stage. Overall, although active-stage T-PLL remain TCR-signaling competent and complete losses of receptor components are rare[19], gaining independence from constant exogenous TCR-input for survival and proliferation might provide a crucial benefit to escape TCR-niche defined growth control[57].

There likely is not one uniform mechanism through which this increasing TCR independence is mediated. Of interest, we detected here a specific deregulation of TCR signaling shared in 4 of our T-PLL patients, namely a loss or downregulation of the phosphatase *PTPRC* (CD45). Progression-related CD45 loss had been described already in single T-PLL patients[58,59]. Apart from its role as a negative regulator of kinase signaling, CD45 has been attributed TCR-signal enhancing functions[60,61], which corroborates our data. The detailed mechanisms by which downregulation of CD45 might affect decreased TCR-signaling activity, are yet to be investigated.

We also unraveled significant downregulations in other immune-related signaling pathways. These signatures were most prominent in active-phase T-PLL cells but were also present in non-neoplastic T-PLL cells of the respective PB samples (micromilieu). Most relevant here were aberrations within the TNFα/NFκB signaling pathway, which included strongly reduced *NFKBIA*, *TNFAIP3*, and *GADD45B* expression, as well as a progressive activation of interferon-α signaling in tumor and non-tumor cells (upregulation of *LY6E*, *IFI44*, *IFI44L*). This aligns with the recognized role of type I interferons in exerting context-dependent immunoregulatory effects[62].

In our analyses, monocytes and dendritic cells were the most informative cell types in the non-tumor compartment, as their relative and absolute numbers increased from indolent to active-stage T-PLL. Most relevant ligands in the communication from T-PLL cells to immune subsets of the tumor microenvironment were upregulated *ANXA1*, as well as homogenously downregulated *CD48*. Both have been implicated to promote an immune-evasive phenotype in different hematologic and solid cancers[44,45,63–65]. In addition, we observed a general pattern of reduced HLA class II gene expression in non-tumor immune cells, potentially reducing cross-presentation of tumor antigens and anti-tumor immune response[66,67]. While computationally based identification of single ligands, as performed in our study, might provide potential targets for future therapy approaches, these results remain preliminary. Further studies incorporating mechanistic in vitro analyses need to decipher the specific role of monocytes and other immune cell subsets in T-PLL.

Adding to the concept of a progressive detachment of T-PLL tumor cells from the tumor microenvironment, our analysis further revealed reduced cell-cell interactions between T-PLL cells and non-tumor immune cells. Escape from immune surveillance is a central step in the progression of many cancers[68–70]. These findings raise the

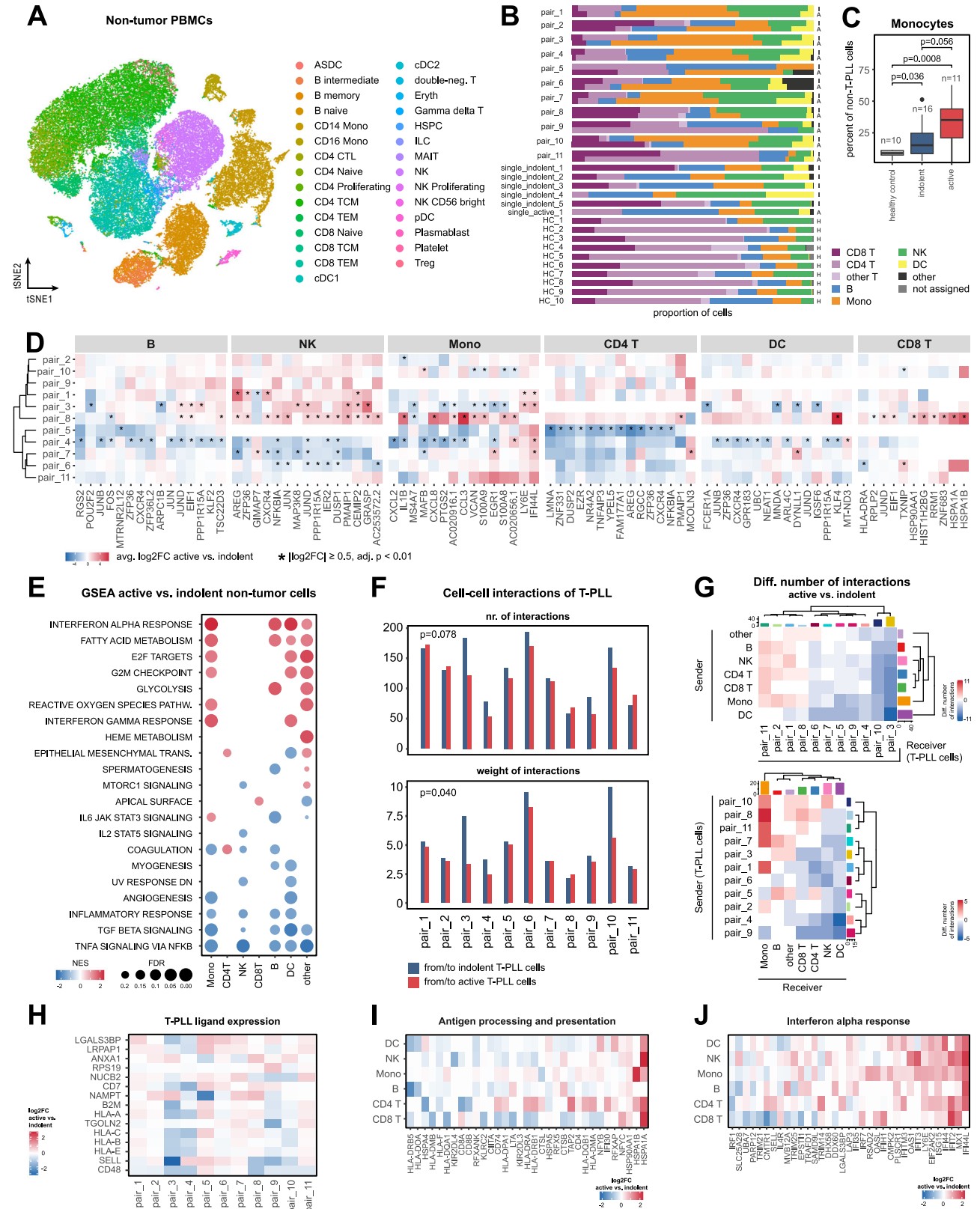

possibility of therapeutic strategies that aim to hijack or reprogram the interaction between T-PLL cells and monocyte or NK-cell populations. Rather than conventional immune checkpoint blockade, which may be futile in T-PLL due to already suppressed expression of *PD-1, PD-L1, CTLA4, LAG3*, and other inhibitory molecules[19], alternative strategies leveraging innate immune effectors such as modulation of the CD47/ SIRPα-axis[71] and blockade of KIR3DL2[72] appear more beneficial.

Extending this to CAR-based cell therapy approaches in T-PLL, it seems warranted to focus on allogeneic cell sources or non-T-cell effectors, given the exhausted T-PLL micromilieu and their lower susceptibility to its immunosuppressive influences. In-depth analyses of the tumor microenvironment are, therefore, upcoming tasks in T-PLL research.

Whole genome sequencing further emphasized the role of ATM as an early and relevant lesion in T-PLL leukemogenesis. Our data

**Fig. 6 | The tumor microenvironment of progressing T-PLL is defined by downregulated immune response pathways. (A)** t-SNE projection of predicted cell type labels in non-tumor cells of T-PLL samples ($n = 28$) and healthy donors ($n = 10$)[73]. **B** Color-coded relative contribution of different non-tumor cell types among samples. Labels on the right y-axis indicate the disease stage of the sample (I: indolent, A: active, H: healthy control). **C** Relative contribution of monocytes to the non-tumor PBMCs of healthy controls ($n = 10$), indolent ($n = 16$), and active-stage ($n = 11$) T-PLL samples (MWW, two-sided). Box plots: center: 50th percentile, box bounds: 25th and 75th percentiles (IQR), whiskers: smallest and largest observations within 1.5×IQR of the box. **D** Differential expression heatmap of recurrently deregulated genes among non-tumor cell types of active vs indolent T-PLL samples. Color indicates the log2 fold-change between active and indolent-stage non-tumor cells (red: upregulated in active T-PLL; white: no change; blue: downregulated in active T-PLL). Significant DEGs (absolute log2 fold-change ≥0.5, adjusted p-value < 0.01, MWW) are indicated with an asterisk. **E** GSEA enrichment of most abundant HALLMARK pathways[36] between active and indolent-stage non-tumor PBMC cell types. Dot colors represent the NES (red: enrichment in active T-PLL, blue: enrichment in indolent T-PLL). FDRs are encoded by size of circles.

**F** Reduced cell-cell interactions between non-tumor PBMCs and active-stage T-PLL cells when compared to indolent T-PLL cells. Interactions derived from CellChat[88]. (paired $t$-test, two-sided). **G** Differential number of cell-cell interactions between non-tumor PBMCs and indolent vs active-stage T-PLL cells stratified by PBMC cell types and incoming vs outgoing interactions. Colors represent differences in interaction numbers between active and indolent-stage T-PLL cells (red: increase in active-stage T-PLL cells, blue: decrease in active-stage T-PLL cells). **H** Gene expression of ligands predicted to modulate the immune microenvironment during T-PLL disease progression. Ligands were inferred using NicheNet based on transcriptomic changes detected in non-tumor immune cells[89]. Colors represent differential gene expression (log2 fold-change) in T-PLL cells comparing active with indolent stages (red: upregulated in active-stage T-PLL cells; white: no change; blue: downregulated in active-stage T-PLL cells). Differential gene expression of most deregulated genes associated with (**I**) antigen processing and presentation, and (**J**) interferon-α response in non-tumor immune cells. Color indicates the log2 fold-change between active and indolent-stage non-tumor cells (red: upregulated in active T-PLL; white: no change; blue: downregulated in active T-PLL). Source data and complete summaries of statistical analyses are provided in the Source Data file.

confirmed the presence of bi-allelic ATM lesions in all analyzed patients, suggesting a two-step pattern, with its early role conferred by its initial lesion (mono-allelic CN loss or mutation) followed by its relevance in advanced stages associated with bi-allelic lesions and/or clonal selection.

In summary, the insights from this study into the previously under-addressed mechanisms of progression in T-PLL, especially around MYC upregulation and alterations of energy metabolism, provide rationales for further translational research in this highly problematic disease.

## Methods

### Patient samples

Primary cells of 17 untreated T-prolymphocytic leukemia (T-PLL) patients from six centers were used in this study. Samples were obtained at different time points during the transition from indolent to active disease stage. Eleven longitudinal sample pairs from T-PLL patients were acquired. The diagnosis of T-PLL was established according to the WHO criteria and T-PLL consensus guidelines[2]. Written informed consent was provided by all patients according to the Declaration of Helsinki. The collection and use of samples have been approved for research purposes by the ethics committee of the University Hospital Cologne (#11–319) and the Medical University of Vienna (1957/2020). Clinical information for all samples is provided in supplementary Data 1. Samples were classified as either indolent or active disease based on recently defined criteria[2]. This stratification is based on multiple parameters, including tumor load (lymphocyte count) in peripheral blood, bone marrow suppression, nodal and extranodal involvement, as well as clinical presentation of T-PLL patients at the time point of sampling.

We did not report on gender. Patient's sex was reported as stated by the patients themselves. We designed the study to maintain a comparable sex-distribution between healthy donors and T-PLL patients. We did not perform analyses stratified by sex because the study was not powered at the sample level to support meaningful sex-based comparisons. Sex is provided as metadata and can be considered in future studies.

A total of 28 different samples were available for this study. After sampling, peripheral blood mononuclear cells (PBMCs) were isolated via density gradient centrifugation (Histopaque, Sigma-Aldrich). Cryo-preserved PBMCs were transferred to the Cologne Center for Genomics (CCG, Cologne) and CeMM-Biomedical Sequencing Facility (BSF, Vienna) for sequencing. Removal of dead cells was accomplished using a Dead Cell Removal Kit (Miltenyi Biotec) or by FACS sorting of DAPI-negative cells. All primary samples were fully consumed during processing and analysis.

### Single-cell RNA-library construction and sequencing

Single-cell suspensions were processed with the Chromium Next GEM Single Cell 3′ Kit v3.1 (10x Genomics) with dual indices according to the manufacturer's instructions, using either one reaction per sample or multiplexing two to three samples with the Cell Multiplexing technology option. In short, cells and the appropriate master mix were loaded on a Chromium Next GEM Chip G and processed on the Chromium Controller to generate Gel Bead-In-Emulsions (GEMs). After the creation of barcoded full-length cDNA by incubation with cell lysate, primers, and a reverse transcription master mix, GEMs were broken, and the pooled fractions were recovered. Silane magnetic beads were used to remove leftover biochemical reagents and primers from the post-GEM reaction mixture. Full-length, barcoded cDNA was then amplified by PCR to generate sufficient mass for library construction. Enzymatic fragmentation and size selection were used to optimize the cDNA amplicon. Library construction was performed, including end repair, A-tailing, adapter ligation, and PCR. The final libraries were quantified (Qubit), pooled, and the pool was quantified using the Peqlab KAPA Library Quantification kit and the Applied Biosystems 7900HT Sequence Detection System. Libraries were sequenced on an Illumina NovaSeq 6000 sequencing instrument with 29 + 89 bp read length. We retained an average sequencing depth of 28,280 reads per cell. Raw reads were aligned to the GRCh38-2020-A reference genome, filtered, and counted using the 10x Genomics Cell Ranger 6.1.2 pipeline at default parameters. UMI count matrices were then further processed using the R package Seurat v4.0.5[34] and R version v4.1 (R Foundation for Statistical Computing).

### Inclusion of healthy donors

To enable temporal analysis of leukemogenesis in T-PLL and facilitate tumor cell identification, we integrated publicly available single-cell RNA-sequencing data of PBMCs from healthy donors (Vu et al., GEO: GSE214284)[73]. Ten donors were selected to closely match the sex and age distribution of our T-PLL cohort (female/male ratio: 1.43 vs 1.5; mean age: 66.9 vs 58 years, T-PLL vs healthy donors). External data sets were generated using compatible technologies (10x Genomics Chromium Single Cell 3′ v3 kit) and processed with Cell Ranger v6 (10x Genomics) to obtain raw feature expression matrices.

### Quality filtering, data integration, dimensionality reduction

The quality of each sample was reviewed individually. We filtered out poor-quality cells keeping cells with at least 250 and not more than 4500 detected features and less than 20% mitochondrial genes. In total, we retained 204,959 single cells (78.6%, supplemental Table 1). UMI counts were normalized to the number of total counts per cell, scaled by a factor of 10,000, and transformed using log1p. Principal

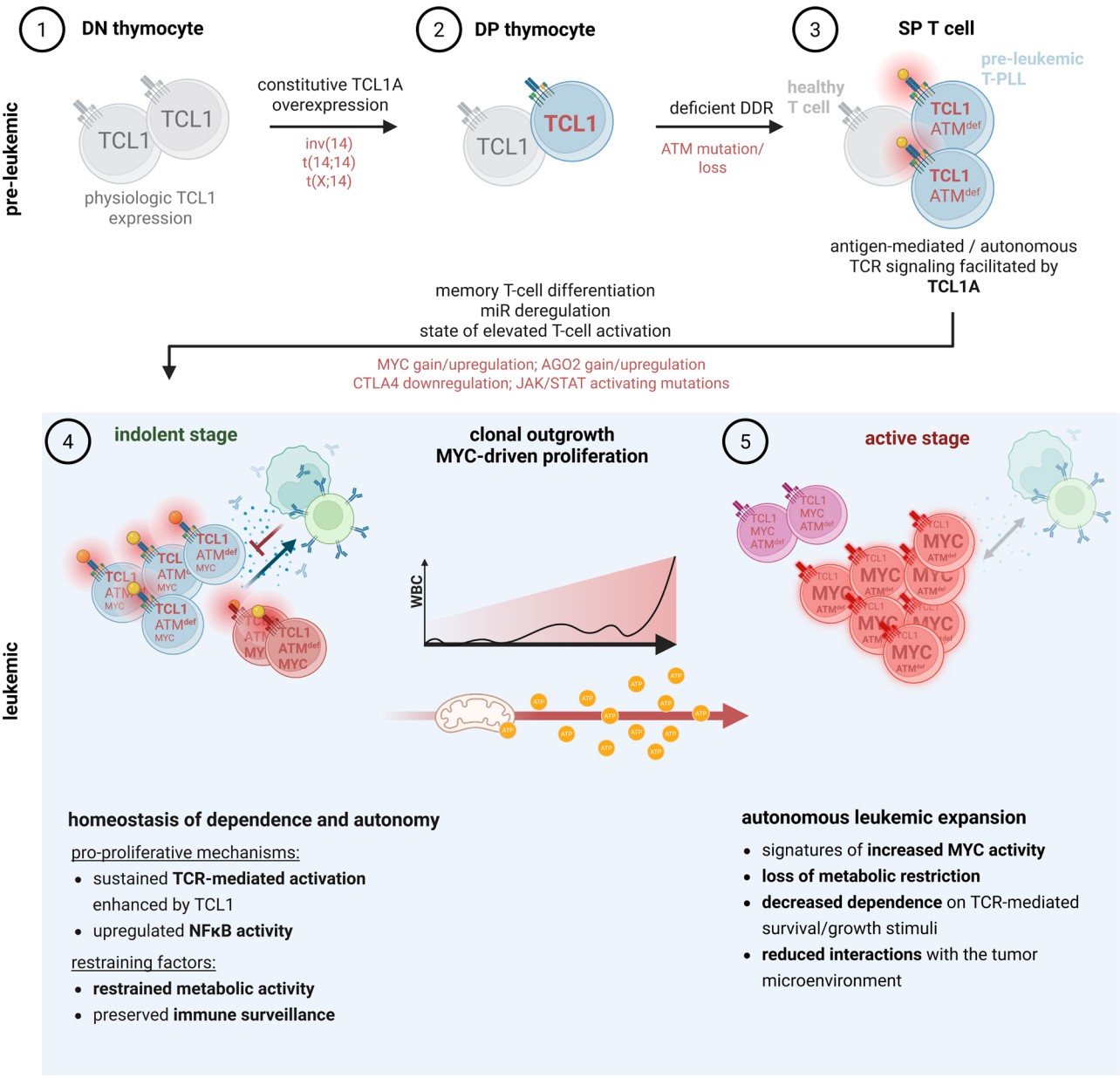

**Fig. 7 | Proposed model of T-PLL leukemogenesis.** Updated model of T-PLL development based on existing genomic and functional data. The proposed trajectory starting from double negative (DN) thymocytes as the initially targeted precursors is illustrated in developmental stages. 1. Physiologically, *TCL1A* expression at the DN stage is silenced upon further thymocyte differentiation towards the single positive (SP) / naïve T-cell stage (gray)[24]. Current disease concepts regard early chromosomal rearrangements that lead to the deregulated expression of the oncogene *TCL1A* or its homologs as the primary lesions in T-PLL leukemogenesis[14,16,19,27]. 2. Additional abrogation of DNA damage response (DDR) pathways, i.e., by deficient ATM, further enables accumulation of relevant genomic lesions in the maturing precursor[13]. 3. Survival and proliferation of pre-leukemic T-PLL cells are strongly driven by TCL1A-enhanced TCR signaling (autonomous and antigen-mediated). During this period the T-PLL precursors acquire additional lesions, including downregulation of negative TCR-signal regulators (e.g., CTLA4, LAG3)[19], activating mutations of JAK/STAT pathway genes[30], and upregulation of AGO2[56] to further facilitate accumulation of an activated memory T-cell pool that has gained advantages through those amplifications of TCR-signal input. Conceivably, other TCL1A effects, e.g., those described in malignant B-cells, such as

aberrant NFκB signaling, abrogated cell-cycle control, and dysregulated epigenetic modulators are at play as well[24,56,97,98]. Upregulation of further oncogenes, such as MYC, propagates outgrowth towards an overt leukemic state (data presented here). 4. During this period of indolent disease, such pro-proliferative mechanisms (also including sustained TCR-pathway activation and NFκB signaling) and restraining factors such as restriction of cellular energy metabolism and immune surveillance by the tumor microenvironment maintain a quasi-homeostatic stage. It is likely that several coexisting subclones harbor different genomic lesions which translate into distinct functional properties. 5. Further clonal evolution would lead to expansion and outgrowth of tumor subclusters that exhibit growth advantages. Here, further enhanced upregulation of MYC signatures and positive selection of MYC- and ATM-altered T-PLL subclusters would propel leukemic expansion. The T-PLL cell progressively detaches from homeostatic control mechanisms, e.g., those imposed by inter-clonal competition[19,57] and, as presented here, by its initial metabolic restriction, or by its tumor microenvironment, and also acquires increasing independence from TCR-related survival signals, all ultimately leading to unrestrained leukemic proliferation during active disease stage. Created in BioRender. Herling, M. (2026) https://BioRender.com/vf2wnou.

component analysis (PCA) was calculated after scaling to reduce the complexity of data. To account for batch effects between healthy-donor derived cells and cells from T-PLL samples and to facilitate subsequent clustering and cell type prediction, batch correction was performed using Harmony with soft parameters (v1.2.3, theta = 0.1)[74]. Dimensionality reduction was performed on all single and integrated data sets using Seurat's implementation of the t-distributed stochastic neighbor embedding algorithm on the top 15 principal components. Subsequent clustering was carried out using the shared nearest neighbor (SNN) modularity optimization of Seurat at a resolution of 1.5.

## Cell type identification

To assign cell type labels to all single cells we followed a semi-supervised approach. First, T-PLL cell clusters were manually identified based on the following criteria: (i) expression of common T-PLL cell markers including CD3D, CD5, CD4, and/or CD8A/CD8B, TCL1A, MYC, and absence of CD19 expression, (ii) independent clustering from other TCL1A-negative T cell clusters in the low-dimensional space of the merged and batch-corrected data set, (iii) expression patterns of TRBC1 and TRBC2 as a surrogate for clonality. Subsequently, non-tumor cells were subjected to an unsupervised prediction of detailed cell labels. For this, we made use of a publicly available CITE-sequencing data set[34] of 162,000 PBMCs from 8 healthy donors that were pre-annotated with a marker panel of 228 antibodies. Count data of non-tumor cells of each sample were normalized using Seurat's SCTransform function and projected into the sPCA space of the reference data set. Cell-type annotation labels were transferred based on the derived anchor sets as previously described[75]. To validate this unsupervised cell type identification, we manually confirmed the predicted cell annotations based on the expression of well-known marker genes. In the analysis of relative cell type distributions, we included samples with at least 50 non-T-PLL cells.

## Differential gene expression analyses

We followed multiple strategies to identify differentially deregulated genes in our data set. Genes significantly deregulated at an adjusted p-value less than 0.05 and an absolute log2 fold-change (log2FC) greater than 0.5 were considered for further analyses. In analyses involving pooled or pseudo-bulk comparisons across individuals, we evaluated the inclusion of age as an additional covariate. However, given the longitudinal structure of the data set, sample age was largely matched by design. Incorporating it as an additional covariate in conjunction with the comparably small sample size of our cohort introduced disproportionate variance attributable to age, which masked disease-related transcriptional changes. We therefore opted to control for age through study design rather than statistical adjustment.

## Pairwise comparison between indolent and active sample-derived T-PLL cells

Differential gene expression was tested between T-PLL cells of the indolent and the active time point of longitudinal sample pairs. We only considered genes expressed in at least 1% of T-PLL cells among all sample pairs. Filtered normalized expression data were submitted to Seurat's FindMarkers function using its default Wilcoxon Rank sum test. Overlaps of differentially expressed genes (DEGs) between sample pairs were illustrated using the R package UpSetR (v1.4.0)[76]. To determine the significance of the number of overlapping genes we compared against a null distribution derived by calculating 10,000 permutations of gene labels.

## Pairwise comparison between shrinking and expanding T-PLL cell clusters

We also investigated gene expression differences between shrinking and expanding T-PLL cell clusters. For this, SNN clustering was performed on a pairwise integrated T-PLL cell data set using a resolution

of 5. Integration anchors were derived using STACAS (v.1.1.0)[77] at default parameters. For each identified cluster, we computed the relative frequency of cells from the indolent and active samples. Clusters were labeled as shrinking or expanding based on whether their relative abundance significantly differed from the expected distribution of cells across time points. Clusters significantly enriched for active-stage T-PLL cells were classified as expanding, while those dominated by indolent-stage T-PLL cells were classified as shrinking. For this, statistical significance was assessed using 10,000 permutations of the observed cell proportions under a null model preserving the overall distribution of indolent and active cells. The analysis was performed using the R package scProportionTest (v0.0.0.9, Robert Policastro) with FDR cutoff of 0.05 and an observed log2 difference in proportion cutoff of 1. This method enabled the identification of clusters significantly enriched in cells from either the indolent or the active time point, reflecting that these clusters were underperforming or overperforming in their growth relative to the other cell clusters. In a second step, normalized gene expression of each cluster was aggregated to create pseudo-bulk expression data. Genes were filtered to features expressed in at least 1% of T-PLL cells of this patient to increase the relevance of identified genes. Differential gene analysis on the generated pseudo-bulks was calculated using the edgeR LRT[78] method implemented in the R package Libra (v1.0.0)[79].

## Comparison of non-tumor cell types between indolent and active T-PLL samples

Similar to the analyses in T-PLL cells, differential gene expression of non-tumor cells between the indolent and active time point was investigated pairwise using Seurat's FindMarkers function on log1p-normalized data. For each cell type and patient, we only considered genes that were expressed in at least 1% of cells.

In addition, we aimed to derive DEGs that were significantly deregulated from indolent to active disease across all patients. In short, normalized single-cell expression values of each cell type and sample were aggregated to build a pseudo-bulk expression data set. Genes were filtered to features present in more than 1% of cells. These pseudo-bulks were then tested for differential gene expression between indolent and active disease over all samples using Libra's edgeR LRT implementation at default values.

## Comparison of T-PLL cells and whole T-PLL PBMC samples to healthy-donor derived T cells

A similar approach was applied to derive deregulated genes between T-PLL cells and healthy-donor derived T cells as well as between whole T-PLL sample PBMCs and healthy-donor derived T cells. Prior to the calculation of DEGs, features were filtered to (i) those being expressed in at least 1% of all cells and that (ii) were detected with at least one read in both data sets. The filtered feature matrix was then subjected to Libra's edgeR LRT function using sex as a confounder, as we observed an influence of sex on the first principal components of gene expression data. Mean expression values were calculated on relative count normalized expression values and then transformed using log1p.

## Pseudotime inference

Linear pseudotime values of T-PLL cells were inferred for each pair separately with the R package SCORPIUS (v1.0.8)[80]. For this analysis, each sample was subsampled to 500 T-PLL cells. We subjected gene expression values of the top 200 differentially expressed genes to the algorithm. Trajectory inference was performed in Spearman-reduced space using k = 10. As SCORPIUS operates unsupervised and therefore does not incorporate a predefined starting or end point, the direction of the pseudotime was oriented after inference such that the mean pseudotime of indolent cells was smaller than the mean pseudotime of active-stage T-PLL cells. This resulted in cell-wise pseudotime scores representing each cell's position on the

trajectory from 0 (indolent T-PLL expression profile) to 1 (active T-PLL expression profile). Inferred pseudotime values were then used to calculate gene importance values of the whole feature expression matrix with SCORPIUS.

In addition to patient-specific pseudotime inference, global approaches incorporating T-PLL cells of many patients are of scientific interest to display the overall transcriptomic shifts during disease progression. In our cohort, the degree of inter-patient transcriptional heterogeneity combined with the low number of commonly deregulated single genes prevented robust alignment and yielded trajectories lacking coherent biological interpretation. As a result, we limited our analyses to patient-specific pseudotime trajectories, which better preserved the temporal structure inherent to the longitudinal sampling design.

## T-PLL bulk mRNA expression data
Validation of single deregulated genes and correlation analysis of global gene expression patterns between our data set and independent T-PLL cohorts were performed using preprocessed expression matrices from two publicly available T-PLL bulk gene expression data sets: (i) Gene expression profiling data performed on Illumina HumanHT-12 v4 BeadChip arrays, which compared PBMCs of 69 T-PLL cases with CD3 + T cells isolated from 10 healthy donors[13] and (ii) mRNA sequencing data that compared PBMCs from 48 T-PLL cases with CD3 + T cells from 6 healthy donors[35].

## Gene set enrichment analyses (GSEA)
All gene set enrichment analyses (GSEAs) were performed using GSEA v4.1.0[81] for Linux on different Molecular Signatures Database (MSigDB) gene set collections (HALLMARK gene sets v7.1[36], Reactome Pathway collection v7.5.1[82], KEGG pathway genesets v7.4[37]). To allow for direct comparison of normalized enrichment scores (NES) between different pairs or cell types, input metrics were subsetted to features that were shared by all data sets prior to calculation. Input genes were then ranked by their fold-change (FC) and provided as input to the GSEA-Preranked method at default parameters. For the GSEA of pseudotime-associated gene importance values, we applied the more conservative classic statistic, which does not involve rank weighting, due to the uncertainty about inter-individual comparability of importance measures. To enhance visualization, enriched Reactome pathways were summarized to their top-level node.

## Gene ontology analysis
Gene ontology analysis was performed using the R package topGO (v2.44.0, Adrian Alexa, Jörg Rahnenführer) on the sub-ontology 'Biological Process'. Overrepresentation of DEGs significant at an adjusted p-value less than 0.01 was tested against the human genome.

## Copy number analyses
Detection of large-scale chromosomal copy number alterations from gene expression data was performed for T-PLL cells of each sample separately. Raw expression matrices were analyzed using the R package inferCNV (v1.9.0, Trinity CTAT Project, Broad Institute of MIT and Harvard). Healthy-donor derived T cells were used as a reference[73]. InferCNV was run on tumor subclusters partitioned by the implemented Leiden algorithm. Predictions were derived with the default Hidden Markov Model (HMM i6). Posterior probabilities were calculated using inferCNV's Bayesian latent mixture model, retaining only those predicted copy number alterations with a probability higher than 0.5. For improved visualization, we summarized identified alterations to the level of chromosome arms, utilizing centromere positions from the UCSC Genome Browser database[83].

## Pathway expression scores
Pathway expression scores were calculated cell-wise using Seurat's AddModuleScore function at default parameters comparing the gene expression of candidate genes to that of randomly sampled control genes exhibiting similar gene expression. Scores were calculated on the entire data set, which comprised tumor cells, non-tumor cells, and healthy-donor derived T cells. Comparisons were conducted between T-PLL cells of different conditions. To facilitate visualization and account for variations in sample sizes, we calculated sample and cluster means. Statistical comparisons of sample means were performed using a permutation test (B = 10,000) to account for the presence of both paired and unpaired samples in the data set.

## Cluster activity scores
Activity scores were calculated on a cluster level using the HALLMARK gene set collection (v7.1)[36] as previously described[84]. To this end, we averaged expression values of the entire disease data set over SNN-derived clusters (resolution = 5) and performed log1p normalization using Seurat. Pseudo-bulk gene expression of each relevant feature was then transformed to percentile ranks comparing all clusters and averaged among gene sets. Finally, scores were shifted by −0.5 to obtain values in the range of [−0.5,0.5] depicting the relative expression strength of a gene set among all clusters.

## Comparison of metabolic transcript expression
Expression of metabolic transcripts was assessed within our single-cell sequencing data set as well as in previously published T-PLL bulk expression data sets. It was defined as the ratio of transcripts that encoded for cell metabolism-associated genes over the expression of all protein-coding transcripts as proposed by Wagner et al.[85]. Features were classified as protein-coding based on the Ensembl 98 genome annotation[86]. Normalized expression values were used to calculate ratios.

## DoRothEA transcription factor activities
Activities of the DoRothEA regulon[43] collection were calculated for each cell separately using the R package decoupleR (v2.2.0)[87]. We used log1p-normalized expression data on a subsampled data set including 1000 cells per sample. Enrichment values were calculated using decoupleR's weighted mean implementation at default parameters. Before plotting, summarized enrichment values were z-transformed to illustrate relative differences in activities.

## Visualization of deregulated T cell receptor pathway components
To visualize the deregulation of T cell receptor (TCR)-signaling components between active and indolent time point T-PLL cells of patient pair_3 we calculated log2 fold-changes of TCR signaling genes as curated by the KEGG pathway database (HSA04660)[37]. Fold-changes were based on log2p1 transformed mean expression values of T-PLL cells.

## Inference of cell-cell interactions
Cell-cell interactions between T-PLL cells and cells of the tumor microenvironment were assessed using the CellChat R package (v2.1.2) and NicheNet (v2.2.0) as previously described[88,89]. In short, we separated non-tumor cells of the indolent and active time points and applied downsampling in order to ensure a comparable number of cells for each cell type, patient, and time point. Immune subsets with fewer than 10 cells were discarded from further analysis. Inference of receptor-ligand interactions between T-PLL tumor cells of each sample and PBMC cells of the respective disease stage was then performed separately using CellChat's human interaction database at default parameters. Next, interactions were filtered to only keep incoming and outgoing signals of T-PLL cells. Differential interactions between indolent and active T-PLL cells of each patient were derived after merging of CellChat objects. Significance of the differential total interaction counts and weights was evaluated using paired Wilcoxon

tests. Finally, differential interaction counts and weights were visualized using CellChat's netVisual_heatmap function.

To identify candidate ligands driving the transcriptional programs observed in T-PLL cells and non-malignant immune cells, we applied NicheNet (v 2.2.0)[89] separately to each longitudinal T-PLL sample pair, using default parameters. We conducted distinct analyses to (i) identify ligands in T-PLL cells potentially mediating the gene expression changes between indolent and active non-tumor cells, and (ii) uncover ligands in T-PLL cells contributing to monocyte-specific transcriptional deregulation. Input gene sets for both analyses were derived using Seurat's FindMarkers function, applying thresholds of adjusted $p \leq 0.05$ and absolute $\log_2$ fold-change $\geq 0.25$. Ligand activity was ranked based on the corrected area under the precision-recall curve (AUPR) as computed by NicheNet.

To improve interpretability and reduce false positive signals in downstream visualizations, we excluded ligands with very low expression (defined as mean $\log2(UMI + 1) < 0.1$). In addition, only ligands expressed in sender populations and for which target genes were sufficiently expressed in receiver populations were considered.

## Whole genome sequencing

A total of 6 T-PLL samples (3 longitudinal sample pairs) were subjected to whole genome sequencing (WGS). DNA was extracted from isolated PBMCs as described in the section on sample processing. Libraries were prepared with the DNA tagmentation-based library preparation kit (Illumina) without PCR, starting with 500 ng gDNA input. Library preparation was followed by clean up and size selection using SPRI beads (Beckman Coulter Genomics). After fluorometric quantification (Qubit, Life Technologies) the libraries were pooled and the pool quantified by real-time PCR using the Peqlab KAPA Library Quantification Kit and the Applied Biosystems 7900HT Sequence Detection System. The library pool was sequenced on an Illumina NovaSeq 6000 sequencing instrument with a paired-end 2x150bp protocol. Samples were sequenced to an average depth of 44.99x. Raw sequencing data were processed following the Genome Analysis Toolkit 4 (GATK4, Broad Institute)[90] best practices for data preprocessing and somatic short variant discovery in tumor-only samples. All tools were run with default parameters using the GATK V4.5.0.0 Docker container. Reference files were obtained from the GATK hg38 resource bundle (v0, Broad Institute).

Reads were aligned to the GRCh38 reference genome using BWA-MEM and preprocessed using Picard's MergeBamAlignment and MarkDuplicates. After base recalibration, raw variant candidates were called with Mutect2 (GATK, Broad Institute) and filtered based on standard parameters using GATK's FilterMutectCalls.

Variant annotation was performed using Funcotator (V4.5.0.0) with the following databases: Gencode 43 CANONICAL, Achilles 110303, ClinVar_VCF 20230717_hg38, Cosmic v98, CosmicFusion v98, CosmicTissue v98, Familial_Cancer_Genes 20110905, Gencode_XHGNC 110_38, Gencode_XRefSeq 110_38, HGNC Jun282023, Oreganno 20160119, Simple_Uniprot 2014_12, dbSNP 9606_b151.

Short variants were retained if they met the following criteria: (i) variant allele frequency (VAF) $\geq 5\%$, (ii) depth $\geq 10$ reads and $\geq 3$ reads supporting the variant, (iii) population allele frequency of $\leq 0.01$, dbSNP MAF $\leq 0.05$, and TOPmed MAF $\leq 0.05$. Predicted functional consequences were derived from Ensembl's VEP release 113[91]. Functional coding variants are provided in supplementary Data 2.

Structural variants and copy numbers were derived from aligned whole genome sequencing data using ClinSV (v1.1.0)[92] at default parameters. High-confidence variants were kept for further analysis.

## Cell culture

Cells were cultured in RPMI-1640 medium including L-glutamine supplemented with 10% FBS and 1% penicillin/streptomycin (100 U/0.1 M) (all Gibco). For TCR crosslinking, $2 \times 10^6$ PBMC/mL were stimulated using plate-bound anti-CD3 (1 µg/mL) and anti-CD28 (2 µg/mL) antibodies (OKT3/CD28.2, BioLegend).

## qRT-PCR

RNA was isolated from PBMCs using Promega ReliaPrep RNA MiniPrep System according to the manufacturer's recommendations. Reverse transcription of total RNA was conducted using the SuperScript VILO cDNA Synthesis Kit (Invitrogen) according to the manual. qRT-PCR was performed on an ABI7500 Fast Real-Time PCR System using the Power SYBR Green PCR Master Mix (both Applied Biosystems). Primers specific to human β-actin were used as a reference for quantification via the 2(-ΔΔCT) method. Primers are listed in the supplements (supplemental Table 2).

## Flow cytometry

To analyze surface expression of T-cell activation markers, $2 \times 10^5$ PBMCs were stained 1:200 with fluorochrome-conjugated antibodies (Supplemental Table 3). Surface expression was measured on a Gallios flow cytometer and analyzed using the Kaluza software (both Beckman Coulter).

## Immunoblots

For immunoblots, 10 µg of whole protein lysates of $1 \times 10^7$ PBMCs were separated via SDS-PAGE and semi-dry transferred to NC membranes using Trans-Blot Turbo Transfer System (BioRad) according to standard techniques[93]. Blocked membranes were stained with primary antibodies and HRP-conjugated secondary antibodies. Antibodies and working concentrations are listed in supplements (Supplemental Table 3). Signals were visualized by Western Bright ECL (Advansta), recorded on autoradiography films (Santa Cruz Biotechnology) using the CAWOMAT 2000 IR (CAWO Solutions), or detected via the ECL Chemostar Imager (Intas). Chemiluminescence was quantified via densitometry using the ImageJ software.

## Glycolysis and oxidative phosphorylation assays

Bioenergetics of glycolysis and mitochondrial respiration were analyzed on the extracellular flux analyzer Seahorse XFe 96 (Agilent, Santa Clara, USA), as previously described[94]. Briefly, one day prior to measurements, Seahorse XFe96 culture plates were coated with Corning™ Cell-Tak Cell and Tissue Adhesive (BD, Franklin Lakes, USA) according to the manufacturer's recommendations. A Seahorse XFe96 cartridge was loaded with XF Calibrant solution and incubated overnight in a CO2-free atmosphere. Also, healthy-donor derived, magnetically isolated T cells as well as patient-derived PBMCs were seeded at a density of $10^6$/mL in RPMI-1640 + 10% FCS + 2 mM L-glutamine + penicillin/streptomycin and cultured in absence/presence of anti-CD2/CD3/CD28 coated activation beads (Miltenyi Biotec, Bergisch-Gladbach, Germany) at a bead-to-cell ratio of 1:2. The next day, cells were harvested from the culture, washed in assay-specific medium according to the manufacturer's recommendations and viable cells were counted. The cells were seeded at a density of $2 \times 10^5$ T cells in 175 µL per well in at least 3 technical replicates. The ports of the Seahorse cartridge were loaded with 25 µL each of 80 mM glucose, 9 µM oligomycin, and 1 M 2DG for the glycolysis stress test and 25 µL of 12.5 µM oligomycin, 13.5 µM FCCP, and of 30 µM antimycin A/rotenone for the mitochondrial stress test. After sensor calibration, assays were run as detailed in the manufacturer's manual by recording ECAR (extracellular acidification rate) and OCR (oxygen consumption rate). Metabolic parameters were obtained from the XF Wave software, calculated using Microsoft Excel, and plotted using R's ggplot package (v3.4.4)[95].

## Statistics and reproducibility

We performed two-tailed statistical tests unless otherwise stated. No statistical method was used to predetermine sample size. No data were excluded from the analyses. The experiments were not randomized.

The investigators were not blinded to allocation during experiments and outcome assessment.

## Reporting summary

Further information on research design is available in the Nature Portfolio Reporting Summary linked to this article.

## Data availability

Filtered feature barcode matrices and raw scRNA data of all T-PLL samples have been deposited at GEO under accession number GSE238130. Data from 10 age- and sex-matched healthy controls (Vu et al., 2024)[73] were downloaded from GEO GSE214284. Bulk gene expression array data (Schrader et al.)[13] were downloaded from GEO GSE107513. Bulk mRNA sequencing data from Braun et al.[35] are available at GEO under GSE318878. The whole-genome sequencing (WGS) data generated in this study contain potentially identifiable germline information from patients and are, therefore, subject to data-protection regulations and restrictions mandated by the informed consent and local ethics approval. Deposition of raw WGS data in a public or broadly accessible controlled-access repository is not covered by the scope of the available patient consent and is, therefore, not permitted under the terms of the relevant informed consent. The consent provided by participants permits long-term storage and scientific use of biospecimens and associated data only under strict governance and access limitations on a case-by-case basis. It explicitly requires pseudonymization while prohibiting measures aimed at re-identification of participants. Access to de-identified raw WGS data is available under restricted access for non-commercial research purposes to qualified researchers whose proposed use is compatible with the original ethics approval and patient consent, and who have obtained approval from their institutional review board or an equivalent ethics committee. Requests for access to WGS data should be directed to the corresponding author (M. Herling, marco.herling@medizin.uni-leipzig.de). Applicants will be asked to provide a short proposal describing the intended use of the data and to sign a data-transfer agreement. Requests will normally be answered within 4 weeks. If access is granted, data will be shared via a secure file-transfer system and will remain available to approved requestors for at least 10 years after publication. Aggregated and processed data from whole-genome sequencing are provided in supplementary Data 2 and in the Source Data file. Source data are provided with this paper.

## Code availability

Analysis code is provided under GitHub [https://github.com/linus235/Single-cell-genomics-in-T-PLL-progression] and is licensed under the MIT License (MIT).[96]

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

## Acknowledgements

We thank D. Saul and C. Fahldieck for their technical assistance in conducting the extracellular flux analyses. In addition, the authors would like to thank the Lille Hospital Tumor Bank for handling, conditioning, and storing patient samples (certification NF 96900-2014/65453-1). This research was funded by the Deutsche Forschungsgemeinschaft (DFG SEQ1176), the EU Transcan-2 consortium 'ERANET-PLL' (01KT1906A), and by the ERAPerMed consortium 'JAK-STAT-TARGET' (ERAPERMED2018-066). A.S. was funded by a scholarship from the German José Carreras Leukemia Foundation (DJCLS 03 F/2016), the French Institute National Du Cancer (INCA-PLBIO2022-072), and Equipe Labellisée LIGUE 2023. S.M. was supported by Cancer Foundation Finland, Academy of Finland and Sigrid Juselius Foundation. T.P. was supported by the Comprehensive Cancer Center Forschungsförderung der Initiative Krebsforschung, MedUni Wien, and by the proceeds of the Krebsforschungslauf and donations to the Initiative Krebsforschung (Comprehensive Cancer Center Vienna-Medical University of Vienna, www.krebsforschungslauf.at). P.B.S. reports grants from Austrian Science Fund (I4154 and I4156). T.M. received a Postdoc scholarship from the Köln Fortune Program (456/2020 and 373/2021). J.B. was supported by the German José-Carreras Leukemia Foundation (DJCLS 01 FN/2024). L.W. and T.B. were supported by a scholarship from the Köln Fortune Program. Additionally, T.B. is funded by the Deutsche Krebshilfe through a Mildred Scheel Nachwuchszentrum scholarship (no. 70113307) and received research grants from the German José-Carreras Leukemia Foundation (DJCLS 10 R_2025), and, together with N. P., from the Sander Stiftung (No. 2023.084.1). N.P. is additionally supported by the DFG (seq-costs in projects, No. PF1028/1-1). T.B. and M.H. are supported by the BMBF/DLR and the SMWK/SAB as parts of the EU ImmuneT-ME consortium (EPPERMED2024-522).

## Author contributions

L.W., D.J., T.M., D.B., T.B., and M.H. designed the computational experiments and analyzed the in-silico generated data. T.P., T.M., M.F., T.G., and K.B. performed sample preparation, sequencing, and primary data curation. Ex-vivo experiments were designed by L.W., D.J., Q.J., T.B., M.B., D.M., and M.H.; D.J., J.B., and M.B. performed the ex-vivo experiments. Statistical analyses were performed by L.W., D.J., and M.B.; D.J., T.P., A.P., S.P., E.J., S.T., T.Z., S.M., E.B., C.H., P.S. and M.H. acquired human tissue samples. S.M., E.B., P.S., M.H., A.S., N.P., and M.H. provided resources. L.W., T.B., and M.H. wrote the manuscript.

## Funding

## Competing interests

T.P. reports being a founder and shareholder of exalt®FlexCo, unrelated to the work. The remaining authors declare no competing interests.

## Additional information

[1]Department I of Internal Medicine, Center for Integrated Oncology Aachen-Bonn-Cologne-Duesseldorf (CIO ABCD), University Hospital Cologne, Cologne, Germany. [2]Center for Molecular Medicine Cologne (CMMC), University of Cologne, Cologne, Germany. [3]Cologne Excellence Cluster on Stress Responses in Aging-Associated Diseases (CECAD), University of Cologne, Cologne, Germany. [4]Department D of Internal Medicine, University Hospital of Muenster, Muenster, Germany. [5]Department of Hematology, Cellular Therapy, Hemostaseology, Infectious Diseases at the University of Leipzig and Cancer Center Central Germany (CCCG) Leipzig-Jena, Leipzig, Germany. [6]Department of Medicine I, Division of Hematology and Hemostaseology, Medical University of Vienna, Vienna, Austria. [7]INSERM 1277-CNRS 9020 UMRS 12, University of Lille, Lille, France. [8]Department of Hematology, Biology and Pathology center, Lille Hospital, Lille, France. [9]Team Lymphoma Immuno-Biology, International Center for Research in Infectious Diseases (CIRI), Lyon, France. [10]Hospices Civils de Lyon, Lyon, France. [11]Claude Bernard Lyon 1 University, Lyon, France. [12]Hematology Research Unit Helsinki, Helsinki University Hospital Comprehensive Cancer Center, Helsinki, Finland. [13]Translational Immunology Research Program and Department of Clinical Chemistry and Hematology, University of Helsinki, Helsinki, Finland. [14]Institute for Molecular Medicine Finland (FIMM), HiLIFE, University of Helsinki, Helsinki, Finland. [15]Department of Hematology and Oncology, University Hospital Magdeburg, Otto-von-Guericke University Magdeburg, Magdeburg, Germany. [16]Health Campus Immunology, Infectiology, and Inflammation (GCI3), Medical Center, Otto-von-Guericke University Magdeburg, Magdeburg, Germany. [17]Cologne Center for Genomics, Medical Faculty, University of Cologne, Cologne, Germany. [18]Medical Department 5—Hematology and Oncology, University Hospital Erlangen, Friedrich-Alexander-University of Erlangen-Nürnberg, Erlangen, Germany. [19]Department of Medical Oncology and Hematology, University Hospital and University of Zurich, Zurich, Switzerland. [20]iCAN Digital Precision Cancer Medicine Flagship, Helsinki, Finland. [21]Haematology Department, Centre Hospitalier Universitaire (CHU) Montpellier, Montpellier, France. [22]Comprehensive Cancer Center Vienna, Vienna General Hospital, Medical University of Vienna, Vienna, Austria. [23]These authors contributed equally: Till Braun, Marco Herling. ✉e-mail: Marco.herling@medizin.uni-leipzig.de

