## [Peer Review File · Nature Communications]

Single-cell genomics highlight MYC-associated metabolic activation and altered cell interactions in T-prolymphocytic leukemia progression

Corresponding Author: Professor Marco Herling

Version 0:

Reviewer comments:

Reviewer #1

(Remarks to the Author)

This is an interesting study comparing the single-cell profile of indolent and active T-prolymphocytic leukemia (T-PLL). The concept of the study is nice but results in limited novel insights due to the small sample size (19 samples), superficial analysis, lack of proper control and validation sets. The conclusions of the study are very confusing as Fig 1D shows that T-PLL is quite homogenous, but it seems heterogeneous based on analysis shown in Fig 2A therefore it seems that authors just implemented a single cell analytical approach without solid rationale behind. The author need to increase sample size and perform validation on independent single-cell data as well as add healthy subjects for reliable tumor cell identification.

Here are major concerns:

- a) The tSNE clustering (Fig1D) of the active and indolent samples after Log Normalization and Integration anchor-based batch correction depict over clustering with minimal segregation of T-PLL. It is not justified why batch correction using the Integration anchor approach was performed. From visualization, data seems to be over-corrected losing intra and interpatient heterogeneity. Authors need to show that batch correction is needed and not cause over-clustering. It seems samples are heterogeneous as Fig2A & B depict large variations in the differentially expressed genes. Please re-evaluate the analysis
- b) In figures 1D & E not clear how many clusters identified, are these clusters patient-specific, Please color UMAP/tSNE with clusters and show contribution of each patient in clusters
- c) Raw data is not available, please make Fastq files available for public access to allow further data analysis. Further, all analytical codes should be also made publicly available to enhance the reproducibility of analysis.
- d) To annotate tumor cells, the study needs to be supported by whole genome or exome analysis for variant analysis, the current CNV analysis is not convincing. The CNV analysis should show the tumor cells have CNV but not normal T cells from the same patients. The analysis needs to be performed through down-sampling of tumor cells due to their significantly higher number than normal T cells.
- e) Further for reliable tumor cell annotation, authors need to perform single-cell analysis on age and gender-matched healthy cells and perform merged analysis for reliable tumor cell annotation.
- f) None of the analyses take into consideration Age, Gender for comparing the proposition cells or gene expression, these confounders need to be adjusted in the analysis.
- g) Not clear if all analysis in Pathways or Differential expression was performed on batch-corrected or non-batch-corrected data, if data has a batch effect then batch-corrected data should be used for all analysis using Limma with confounders
- h) The study has Indolent and Active samples from 7 patients, it will be great to perform trajectory analysis to see which cells clonally expand from Indolent to Active disease transition. How the origin cluster for Pseudotime analysis shown in Figure 2F is decided ?.
- i) The approach for identifying shrinking and expanding clusters is not clear (Fig 2D). The clusters with significant change (mean and SD) from indolent to active phases in each patients should be used for defining shrinking and expanding clusters for downstream analysis.
- j) Cellular Communication analysis on tumor cells that change from indolent to active disease with immune cells might provide some interesting insight into key molecules critical for disease progression. The analysis is lacking in the current study.

k) Most of the gene targets, pathways, and gene set boxplots shown in the manuscript are not able to achieve significance this might be due to the limited sample size. The significance testing outcomes should be added to each figure Fig 3A, 4A, 4D,6F, 6G etc. A higher sample size is needed to make sure that results are not false positive. The authors should also add another single cell dataset to validate the findings from the preliminary analysis.

l) The authors show that single-cell analysis results related to MYC targets, oxidative phosphorylation, and glycolysis seem similar to results from previous bulk RNA studies, so it is not evident what novel insights single-cell profiling is providing. Therefore, further deeper analysis of larger single-cell data might provide novel insight into cellular interaction/communication that is involved in the progression of indolent to active disease. The current findings seem very generalized and associated with the progression of any cancer not only T-PLL.

Reviewer #2

(Remarks to the Author)

In this study, the authors investigated the potential mechanisms underlying the progression of T-PLL from an indolent phase to an active phase. They identified distinctive features in active-stage T-PLL, including upregulated MYC pathways, increased metabolic activity, downregulation of immune-response pathways and decreased interaction with cells in microenvironment. The study is valuable, well designed and the results are supportive of the conclusions. Couple comments:

- 1, The authors meticulously detailed the alterations at the RNA level. Have the authors had an opportunity to examine genetic changes at the mutational and karyotyping levels? Specifically, are there any additional mutations and/or cytogenetic changes identified in association with the progression of the disease from an indolent to an active state?
- 2, JAK/STAT pathway plays an important role in T-PLL. Is there any change in this pathway (additional mutation involving this pathway or overexpression) during disease progression from indolent to active?
- 3, In the discussion, there is an opportunity for the authors to further elaborate on their novel findings and explore potential therapeutic implications, particularly considering the poor prognosis associated with T-PLL. The authors mentioned MYC as potential therapeutic target, how about potential targets in metabolic pathways? This study showed downregulation of immune-related signaling pathways. Any potential role of immunotherapy? The findings of tumor-associated monocytes are also interesting, and any potential treatment targeting tumor associated monocytes/macrophage and other cells in the microenvironment?
4. The discovery of downregulation of PTPRC (CD45) is intriguing. In my daily practice, I have encountered several cases of T-PLL exhibiting decreased to negative CD45 expression as assessed by flow cytometry.
5. From Figure 3G, we can see that MYC is active and its expression is increased in indolent phase. The author may want to briefly mention and discuss it.
6. Can the author comment on ATM mutation in T-PLL? Can this be a potential therapeutic target? ATM mutations in T-PLL are very common. In contrast, they are extremely rare in other T-cell lymphomas/leukemias. How is the authors' experience? How often do you see ATM mutations in other T cell neoplasms?
7. I may miss this information. Did all T-PLL cases in this study carry TCL1 rearrangement? Any cases with t(X;14)?

Reviewer #3

(Remarks to the Author)

General comments:

The authors performed single-cell RNA-seq of 7 longitudinally paired and 5 single time-point derived T-PLL samples to study the molecular changes underlying the evolution from indolent to active T-PLL. Based on the fact that T-PLL is a rare cancer, the study of 7 paired cases is of interest, but has limitations. Overall, my feeling is that the study is performed very well, technically there are no concerns and data analysis seems very robust, but due to the low number of cases and the variability from case to case, there are no highly novel conclusions: it is well known that TCL1A is implicated in the pathogenesis of T-PLL and also that increased MYC expression caused by MYC amplification is a common feature of T-PLL.

The analysis of differentially expressed genes (DEGs) in the paired samples is not very informative and seems to indicate that each case has a rather unique set of DEGs.

Since MYC is very important in the pathogenesis of T-PLL, it is not surprising that the most uniform shifts of gene expression across all indolent/active sample pairs were detected in GSEA-confirmed functional gene sets associated with cell metabolism and cell-cycle regulation, which are very much controlled by MYC, as previously illustrated for T-ALL and other T-cell malignancies.

Interestingly, the authors investigated differential gene expression between T-PLL cell clusters that expanded compared to those that retracted over time and identified in this way a strong enrichment of gene sets related to cell cycle, energy metabolism, and the transcription factor MYC in the expanding T-PLL cell clusters. This is an interesting analysis, and one of the strongest parts of the study. However, I am not sure how this was performed as there were no clear clusters shown in the malignant cells based on the analysis of the single-cell data shown in figure 1. How were these clusters identified? I could not find a figure clearly illustrating these clusters.

Specific major comments:

Figure 1:

On figure 1D/E the single cell data has been integrated removing all patient specific information, but at the same time also removing some of the indolent-active information. Indeed, on figure 1E we see a gradient from indolent to active, but one might expect a clearer separation between the two phases. It would be interesting to see the data before this data integration where the single cells cluster according to the patient and the timepoint. The normal cells of all patients still should cluster together in this plot.

Why does the patient-specific data needs to be removed in this analysis? All subsequent comparisons are on patient level, which means that comparing the indolent with the active samples of the same patient already filters out patient specific information.

In figure 1G the different clusters are annotated. For me it seems weird that the CD4 T-cells cluster more closely to the B-cells than to the CD8 T-cells. Could this be due to the data integration?

On line 185 it is stated that a high intra- and inter-sample heterogeneity is observed. This is not clear from this figure as all samples cluster together on the t-SNE plot. In supplementary figure 1B, we see clustering based on the patients, which would indicate that there is not a lot of intra-tumor heterogeneity. Please elaborate.

In the differential analyses the authors used data on normal cells from an independent study. Why was it not possible to use the normal cells that were sequenced in the current study? How did the authors deal with possible batch effects? Please elaborate.

Figure 2:

In figure 2G a GSEA is performed on the pseudotime correlated genes. How were these genes determined? How many genes are included? Is the number large enough to perform GSEA?

Does the pseudotime correlate with the indolent/active timepoints? Maybe add a tSNE/UMAP plot with the pseudotime (in the supplementary)?

What is the added value of this analysis in comparison to the analyses on the indolent-vs-active and the expanding-vs-shrinking? Please explain.

Figure 5:

This is well presented, but does not provide a lot of new information as most of the data shown here has been reported previously. Data on CTLA4 were reported in Tian, S. et al. Sci Rep 11, 8318 (2021).

Figure 6:

In figure 6G/H the cell-cell interactions are investigated. However, only the number of interactions are reported here. It would be interesting to see what kind of interactions are happening (and disappearing).

While the changes in cell-cell interactions determined by single-cell RNA-seq analysis are of interest, these remain predicted (and not confirmed) and is not very convincing. Conclusions are limited.

Minor comments:

- In figure 5A the heatmap is clustered both on the gene expression and the samples. As this figure wants to indicate a difference between the indolent and the active timepoints (line 328) it would be interesting to see the samples clustered according to their timepoint.

- In figure 6D, there is a character ']' missing in the legend.

Reviewer #4

(Remarks to the Author)

Version 1:

Reviewer comments:

Reviewer #2

(Remarks to the Author)

All of my comments are addressed thoroughly and appropriately.

(Remarks on code availability)

Reviewer #3

(Remarks to the Author)

The authors have responded to all comments and questions. I have no further remarks.

(Remarks on code availability)

Reviewer #4

(Remarks to the Author)

(Remarks on code availability)

point-by-point replies to reviewers:

Reviewer #1

This is an interesting study comparing the single-cell profile of indolent and active T-prolymphocytic leukemia (T-PLL). The concept of the study is nice but results in limited novel insights due to the small sample size (19 samples), superficial analysis, lack of proper control and validation sets. The conclusions of the study are very confusing as Fig 1D shows that T-PLL is quite homogenous, but it seems heterogeneous based on analysis shown in Fig 2A therefore it seems that authors just implemented a single cell analytical approach without solid rationale behind. The author need to increase sample size and perform validation on independent single-cell data as well as add healthy subjects for reliable tumor cell identification.

REPLY:

We thank Reviewer #1 for the thoughtful and constructive evaluation of our manuscript. We value the reviewer's expertise and the detailed consideration of our work, and we are grateful for the opportunity to address the concerns raised. Our rationale was to decipher molecular changes during T-PLL evolution, which the exceptionally rare samples from inactive disease stages and their pairing with active phases uniquely allow, and for which a single-cell approach is ideally suited.

T-PLL itself is an exceptionally rare leukemia, with an incidence of ~1 per million individuals per year. Indolent cases, the focus of our study, constitute only about 20% of all diagnoses, and longitudinally paired, treatment-naïve samples of indolent and active phases are even more scarce. To put this into perspective: since 2008, a dedicated mono-centric collection in Cologne combined with a national registry effort has led to the banking of 496 samples from 252 T-PLL patients, among which only very few inactive cases and an even smaller number of longitudinally matched, untreated, pairs could be identified. The cohort we assembled therefore represents an unparalleled collection that has required years of concerted effort.

At the international level, we reached out to major centers and reference collections to compile the cohort underlying the first submission of this work. In our revision, we have gone further: by contacting more than 15 additional research facilities worldwide over the past 18 months, we were able to expand our dataset with four additional indolent/active pairs and one more indolent case. Together with new analyses and the inclusion of additional age- and gender-matched healthy controls, this substantially strengthens the robustness and impact of our findings.

Implementing a single-cell approach allowed us to resolve dynamic tumor subpopulations and to characterize the tumor microenvironment, both of which would be obscured in bulk RNA-seq due to signal averaging and the low abundance of non-tumor cells. To our knowledge single-cell resolved genomic analyses of T-PLL have not been performed to date. Our longitudinal sampling further enabled a chronologically resolved analysis of both malignant and non-malignant compartments, providing novel insight into disease progression. Below, we provide a point-by-point response to each of the reviewer's comments.

Here are major concerns:

1) The tSNE clustering (Fig1D) of the active and indolent samples after Log Normalization and Integration anchor-based batch correction depict over clustering with minimal segregation of T-PLL. It is not justified why batch correction using the Integration anchor approach was performed. From visualization, data seems to be over-corrected losing intra and interpatient heterogeneity. Authors need to show that batch correction is needed and not cause over-clustering. It seems samples are heterogeneous as Fig2A & B depict large variations in the differentially expressed genes. Please re-evaluate the analysis

REPLY: We thank the reviewer for this comment. We agree that the original approach leveraging anchor-based integration for the purpose of clustering and cell type identification (Fig. 1D-G) may have inadvertently conveyed

a misleading impression of homogeneity among T-PLL samples, and we appreciate the opportunity to clarify our rationale and improve this aspect of the analysis.

In the revised manuscript, we have restructured our approach of data integration and visualization. Specifically, we removed anchor-based integration in the initial visualization step as we did not observe substantial batch effects across our T-PLL patients (**rebuttal Figure 1**). Importantly, non-tumor cells from all T-PLL patients clustered homogeneously and in a cell-type specific manner, whereas T-PLL cells clustered separately and in a patient-specific manner. This pattern suggests that the observed variability is driven by biological differences of T-PLL cells rather than by technical confounders. However, as we did identify small, but persistent, batch effects between the non-tumor compartment of T-PLL patients and PBCMs of independently sequenced healthy controls,¹ we decided to apply the Harmony² algorithm with soft parameters ($\theta = 0.1$) to account for this confounder without overriding patient-specific expression signatures. In line with the reviewer's suggestion, we now provide tSNE clustering performed on merged normalized datasets and believe that these better represent the heterogeneity among T-PLL cells observed throughout our study.

Rebuttal Figure 1

Rebuttal Figure 1. Overview on non-batch corrected gene expression data and possible confounders

tSNE representation of T-PLL samples from both indolent and active disease and external healthy controls prior to batch correction using Harmony. Subpanels illustrate the distribution of different biological and technical parameters. Within T-PLL samples, malignant cells cluster in a strongly patient-specific manner, while non-tumor cells group by cell type. Before batch correction, non-tumor cells from T-PLL patients are distinct from healthy-donor PBMCs. For example, B cells (top row, blue) form two separate clusters, one comprising B cells from T-PLL patients and the other from healthy donors.

To increase transparency for the reader regarding potential batch effects within our data and the performed batch correction, we included a new supplementary figure (**supplemental Figure 1B**) comparing different potential confounders in tSNE embeddings without batch correction. We demonstrate that our revised approach retains

inter-patient heterogeneity with regard to T-PLL cell gene expression while still minimizing confounding batch effects between T-PLL samples and healthy-donor derived cells.

Importantly, consistent with the first version of the manuscript, all downstream analyses, including differential expression, pathway analysis, and pseudo-time inference, were performed on non-integrated, normalized expression matrices, ensuring that no biological signal was lost due to batch correction.

2) In figures 1D & E not clear how many clusters identified, are these clusters patient-specific, Please color UMAP/tSNE with clusters and show contribution of each patient in clusters

REPLY: We acknowledge the need for greater clarity in the presentation of our clustering results. In the revised manuscript, we now provide a supplementary figure (**supplemental Figure 1D, rebuttal Figure 2**) showing all identified clusters used for cell type identification and included a figure panel quantifying the contribution of each patient to each cluster.

Rebuttal Figure 2

clusters used for celltype identification

Rebuttal Figure 2. Identified clusters and distribution of clusters among patients

Left panel: Identified clusters as used for cell type identification. Clusters were derived using Seurat's shared nearest neighbor (SNN) modularity optimization at a resolution of 1.5 on batch-corrected PCA space. Right panel: Stacked bar plot representing the relative contribution of T-PLL patients and healthy controls (HC) to each identified cell cluster (cluster # at x-axis). Bars are colored by individual patients (see color legend), with some clusters displaying patient-specific enrichment, while clusters corresponding to non-tumor cell populations include contributions from multiple patients. Clusters are ordered by cellular composition diversity, from those with highly mixed contributions across individuals (left) to those dominated by a single patient (right).

3) Raw data is not available, please make Fastq files available for public access to allow further data analysis. Further, all analytical codes should be also made publicly available to enhance the reproducibility of analysis.

REPLY: We fully support the principles of reproducibility and transparency. In response, we have deposited all raw scRNA-seq data to GEO under accession number GSE238130 (Reviewer access token: **yzclceigzdmzzmt**). The new whole-genome sequencing (WGS) data will be made available upon reasonable request to us and in accordance with German regulations of data protection. All scripts, workflows, and pre-processing steps used in the analysis are now available via a publicly accessible GitHub repository (<https://github.com/linus235/Single-cell-genomics-in-T-PLL-progression>) to facilitate reanalysis.

4) To annotate tumor cells, the study needs to be supported by whole genome or exome analysis for variant analysis, the current CNV analysis is not convincing. The CNV analysis should show the tumor cells have CNV but not normal T cells from the same patients. The analysis needs to be performed through down-sampling of tumor cells due to their significantly higher number than normal T cells.

REPLY: We thank the reviewer for this important point. We fully acknowledge the limitations of expression-based copy-number inference and appreciate the recommendation to validate our findings using orthogonal genomic data. In response, we have now incorporated new matched WGS data from six samples (three T-PLL patients, each with paired indolent and active-stage samples). This WGS was performed on bulk PBMCs, which predominantly consisted of leukemic T-PLL cells (median tumor fraction: 92.97%), with only minimal contamination by residual non-malignant T cells (median: 1.7%).

We also understand and agree with the reviewer's suggestion that using patient-matched normal T cells would be preferable for defining tumor-specific CNV profiles. In line with this, we attempted to use pooled non-malignant T cells from within the T-PLL samples as an internal reference for transcriptome-based CNV inference. However, due to their very limited number and variable transcriptomic quality, this approach produced low concordance with the WGS-derived CNV profiles, particularly in key lesions (see **rebuttal Figure 3**).

Given these technical limitations, we retained the use of healthy-donor T cells as the reference population for CNV inference in the main manuscript. This strategy now benefits from the independent validation through the newly generated WGS data, which substantially strengthens the reliability of our CNV analyses.

Rebuttal Figure 3

Rebuttal Figure 3. Copy number prediction leveraging health-donor derived T cells as a control shows high correlation to WGS-based copy number inference.

Dot plots showing the correlation between predicted copy numbers for T-PLL key genomic alterations comparing different technical approaches. Top: WGS-derived copy numbers for *MYC* and *ATM* are compared to copy numbers inferred from scRNA-expression data using healthy-donor T cells as controls. Predicted copy numbers show a high correlation between both methods (*MYC*: $r=0.9816$, $p=0.0005$; *ATM*: $r=0.9817$, $p=0.0005$, Pearson correlation) affirming the validity of the approach of utilizing healthy-donor T cells. Bottom: Copy numbers for *MYC* and *ATM* as predicted from WGS data are compared to scRNA-expression derived copy numbers using non-tumor T cells of T-PLL patients as controls. Here, correlation between both methods appears weaker as compared to the approach using healthy-donor T cells as controls (*MYC*: $r=0.8284$, $p=0.0416$; *ATM*: $r=0.7419$, $p=0.0913$, Pearson correlation).

5) Further for reliable tumor cell annotation, authors need to perform single-cell analysis on age and gender-matched healthy cells and perform merged analysis for reliable tumor cell annotation.

REPLY: We fully agree that matching for age and sex is crucial to control for potential confounders in scRNA-seq data. In response to this valuable comment, we have now included 10 additional PBMC samples from healthy donors, carefully matched to the T-PLL cohort in terms of age and sex (female/male ratio: 1.43 vs. 1.5; mean age: 66.9 vs. 58 years, T-PLL vs. healthy donors).¹ These controls were incorporated into the tumor cell annotation pipeline and used as a reference for CNV detection, thereby enhancing biological validity. Importantly, in contrast to many other malignancies, tumor cell identification in T-PLL is comparatively straightforward due to the high tumor cell burden, strong genomic and transcriptomic signatures, including the frequent distinguishing overexpression of *TCL1A*, a well-established oncogene present in the majority (>90%) of T-PLL and absent in non-malignant T cells.³

6) None of the analyses take into consideration Age, Gender for comparing the proposition cells or gene expression, these confounders need to be adjusted in the analysis.

REPLY: Thank you for raising this important point. Because the core of our study involves intra-individual comparisons of indolent and active samples, age and sex are inherently matched for the majority of our analyses. In gene expression analyses of pseudo-bulk samples, we adjusted for sex as suggested by the reviewer (**supplemental Figure 2**).

Notably, we did not observe any dominant age-related patterns in our gene expression data (**rebuttal Figure 4**). Given our naturally small cohort size, including multiple covariates in statistical models would be statistically impractical, risking overfitting, and substantially reducing power to detect true biological signals. We, therefore, believe that controlling for age at the design level, rather than adjusting for it post hoc, is the most appropriate approach in this context. We have added a detailed explanation of this decision to the revised Supplemental Methods section.

Rebuttal Figure 4

Rebuttal Figure 4. PCA of gene expression values comparing active-stage T-PLL cells and healthy-donor derived T cells

(A) Dot plot showing the first two principal components from PCA of gene expression in active-stage T-PLL cells and healthy-donor derived T cells. Colors indicate the sex of the respective sample. Among T-PLL samples there is a relevant association of patient's sex and PC1. (B) PCA as performed in A. Text and point colors represent the age [years] of patients and healthy controls at the timepoint of sampling. There is no substantial effect of age on the first two PCA components.

7) Not clear if all analysis in Pathways or Differential expression was performed on batch-corrected or non-batch-corrected data, if data has a batch effect then batch-corrected data should be used for all analysis using Limma with confounders

REPLY: We thank the reviewer for this request for clarification. We confirm that all differential gene expression and pathway enrichment analyses were conducted on normalized, non-integrated count matrices as commonly suggested. Batch correction was applied for the purposes of visualization, clustering, and cell type identification, to facilitate interpretation. This approach ensures that the underlying biological variation is preserved in statistical testing and that technical artifacts do not bias our conclusions.

8) The study has Indolent and Active samples from 7 patients, it will be great to perform trajectory analysis to see which cells clonally expand from Indolent to Active disease transition. How the origin cluster for Pseudotime analysis shown in Figure 2F is decided ?

REPLY: We agree that the question of clonal evolution and its modeling in pseudotime is of central importance. In this study, we employed the SCORPIUS⁴ algorithm to infer pseudotime separately for each patient. SCORPIUS does not require a predefined origin cluster, but does independently predict a linear undirected pseudotime across all cells. For all patients, the trajectory was later oriented such that indolent cells had lower pseudotime values. This approach yielded robust and interpretable trajectories across individuals.

We also attempted to derive a unified pseudotime across all samples using Monocle3⁵ and Slingshot.⁶ However, due to substantial inter-patient heterogeneity and the absence of consistently differentially expressed genes across patients, these global models failed to produce biologically meaningful and stable trajectories. The resulting outputs were highly sensitive to the choice of hyperparameters. We now clarify in the Supplementary Methods why we did not include a shared pseudotime analysis. We agree that the development of a pan-patient pseudotime model of disease evolution remains an important goal for future studies (however, being limited by the need of larger cohorts). In response, we chose to move the pairwise pseudotime analyses to the supplemental data.

9) The approach for identifying shrinking and expanding clusters is not clear (Fig 2D). The clusters with significant change (mean and SD) from indolent to active phases in each patients should be used for defining shrinking and expanding clusters for downstream analysis.

REPLY: We apologize for the lack of detail in our original description. Expanding and shrinking clusters were indeed defined as suggested by the reviewer for each patient separately. Specifically, we calculated the relative contribution of T-PLL cells from each time point to each cluster and retained those clusters that were significantly enriched for cells from a particular time point. We expanded the corresponding passages in the Supplemental Methods as well as in the figure legends (**Figure 2**) to enhance clarity and to improve understanding. Additionally, all analysis codes have now been made publicly available within the GitHub repository.

10) Cellular Communication analysis on tumor cells that change from indolent to active disease with immune cells might provide some interesting insight into key molecules critical for disease progression. The analysis is lacking in the current study.

REPLY: We thank the reviewer for this excellent suggestion. As originally included in our manuscript, we had already performed cell-cell communication modeling using CellChat,⁷ which revealed a global reduction in intercellular signaling in active T-PLL compared to the indolent stage. In line with the reviewer's recommendation,

we have now expanded and refined this analysis to better capture cell-cell interactions relevant to disease progression.

In addition to the CellChat framework, we applied NicheNet⁸ to further dissect ligand-receptor dynamics and their downstream effects on gene expression in non-malignant immune cells (**rebuttal Figure 5**). Despite considerable inter-patient variability, this extended analysis revealed relevant and recurring patterns across samples. Specifically, we analyzed ligand expression profiles on the T-PLL cells themselves to better understand how leukemic T cells may actively shape their immune surroundings and to explain the transcriptomic dynamics observed within the tumor microenvironment.

In this context, two ligands stood out as consistently deregulated in the active disease stage: Annexin A1 (*ANXA1*) and *CD48*. First, *ANXA1* was upregulated in T-PLL cells during disease progression. As an endogenous anti-inflammatory mediator with known immunomodulatory properties, *ANXA1* can suppress dendritic cell maturation and promote the resolution of immune responses.^{9,10} Its increased expression in active-stage T-PLL is, therefore, consistent with an actively immunosuppressive phenotype, possibly contributing to reduced immunogenicity in the surrounding tumor microenvironment (immune escape). Second, we observed homogenous downregulation of *CD48* in T-PLL cells of the active stage. *CD48* is a co-stimulatory ligand for 2B4 (*CD244*), which is expressed on NK cells and cytotoxic T lymphocytes. This interaction typically promotes immune synapse formation and cytolytic activity.¹¹ Loss of *CD48* expression has been previously described as a mechanism of NK-cell immune evasion in various hematologic malignancies.¹²⁻¹⁴ Its downregulation in T-PLL cells may, therefore, reduce their recognition and killing by NK cells, further supporting that active mechanisms of immune escape are at play in active T-PLL.

Furthermore, building on our initial identification of genes deregulated in T-PLL cells that influence the tumor microenvironment, we have now extended our analyses to the reverse direction: we systematically investigated genes deregulated within microenvironmental cell populations that may, in turn, shape T-PLL biology. These analyses revealed candidate pathways contributing to the diminished cell-cell interactions and reduced immune responses we had observed.

Here, we recurrently identified progressive downregulation of genes associated with antigen processing and presentation among all analyzed immune cell subsets during disease progression. While T-PLL tumor cells presented mostly downregulation of HLA class I genes, non-tumor immune cells revealed reduced expression of HLA class II genes. This global pattern might potentially contribute to a reduction of antigen cross presentation ultimately reducing the anti-tumor immune response. Similar patterns have been identified in other entities.¹⁵⁻¹⁷ In addition, we observed strong upregulation of *IFI44*, *IFI44L* and *LY6E* in immune cells of active-stage T-PLL samples conveying the general upregulation of interferon-alpha signaling during disease progression observed in our data. While interferon signaling has traditionally been attributed tumor suppressive functions, recent studies have increasingly demonstrated its role also in the negative regulation of anti-tumor immunity, mainly through induction of immunosuppressive cytokines and promotion of tumor associated macrophages.¹⁸ Interferon-inducible *LY6E* has widely been accepted as a central regulator of the anti-tumor immune response associated with CD8+ T cell exclusion and resistance to immunotherapy.^{19,20}

We have integrated these new findings into **Figure 6** and added a new supplementary figure (**supplemental Figure 13**) summarizing the most relevant ligands and patterns of transcriptional reprogramming across (anti-tumor) immune cells. In the revised Results, Supplementary Methods, and Discussion sections, we now provide a more comprehensive overview of how shifts in cell-cell communication may contribute to the evolving immune microenvironment in T-PLL. We certainly acknowledge in that respect the technical limitations inherent to scRNA-seq data, particularly with regard to low-abundance cytokines and immune-signaling molecules, and the necessity of subsequent *in vitro* and *in vivo* functional studies outside the scope of this study.

Rebuttal Figure 5

Rebuttal Figure 5. Recurrently deregulated ligands in the (immunosuppressive) T-PLL-immune cell interaction

(A) Gene expression of ligands predicted to modulate the immune microenvironment during T-PLL disease progression. Ligands were inferred using NicheNet based on transcriptomic changes detected in non-tumor immune cells. Colors represent differential gene expression (\log_2 fold change) in T-PLL cells comparing active and indolent stages (red: upregulated in active-stage T-PLL cells; white: no change in gene expression; blue: downregulated in active-stage T-PLL cells). LGALS3BP, ANXA1, and CD48 were among the most recurrently identified and deregulated effectors. (B) Differential gene expression of most deregulated genes associated with antigen processing and presentation in non-tumor immune cells. There was a strong downregulation of HLA class II genes across PBMC subtypes in active-stage T-PLL while genes coding for heat shock proteins were uniformly upregulated. Color indicates the \log_2 fold-change between active and indolent-stage non-tumor cells (red: upregulated in active-stage T-PLL; white: no change in gene expression; blue: downregulated in active-stage T-PLL). (C) Gene expression shifts of interferon-alpha response-associated genes during disease activation in non-tumor PBMCs. There was a strong upregulation of type I interferon signaling among all immune subsets. Color indicates the \log_2 fold-change between active and indolent-stage non-tumor cells (red: upregulated in active-stage T-PLL; white: no change in gene expression; blue: downregulated in active-stage T-PLL).

11) Most of the gene targets, pathways, and gene set boxplots shown in the manuscript are not able to achieve significance this might be due to the limited sample size. The significance testing outcomes should be added to each figure Fig 3A, 4A, 4D,6F, 6G etc. A higher sample size is needed to make sure that results are not false positive. The authors should also add another single cell dataset to validate the findings from the preliminary analysis.

REPLY: Thank you for this valuable point. We acknowledge that sample size is a limitation. T-PLL is an exceptionally rare disease, with an incidence of approximately 1 per million individuals per year. Indolent cases, on which our analysis is focused, constitute only around 20% of all T-PLL diagnoses. Furthermore, our study specifically sought longitudinally paired and treatment-naïve primary samples, which significantly restricts availability, but simultaneously underscores the uniqueness and value of our cohort. To address this concern, we have significantly expanded our cohort by contacting more than 15 research facilities all over the world. This allowed us to include four new indolent/active sample pairs and one additional indolent case. In addition, we added previously missing formal statistical testing for all key analyses, as requested by the reviewer. We further strengthened our data as our key findings proved to be consistent and statistically significant also in this expanded cohort. This consistence was very assuring in terms of robustness of our findings. A representative exemplary comparison of key figure panels, both prior and post inclusion of the additional samples (for these revisions), can be derived from **rebuttal Figure 6**.

While no comparable single-cell RNA-seq dataset currently exists for T-PLL, and to our knowledge, no prior gene expression analyses have included annotated indolent-stage T-PLL, we did perform validation of single deregulated genes and pathways, where possible, using independent bulk RNA-seq data from two publicly available T-PLL cohorts^{21,22} (e.g., see **Figure 3D** and **Figure 4D**). Although these datasets do not capture the early indolent phase, these analyses revealed strong concordance with our scRNA results for these individual genes and pathways.

Rebuttal Figure 6

A GSEA expanding vs shrinking T-PLL clusters

B MYC targets V1

C Oxidative phosphorylation

Rebuttal Figure 6. Comparison of key analyses prior to (original submission) and after expansion (revision) of the T-PLL cohort size. (A) Gene set enrichment analysis (GSEA) calculated on differentially expressed genes between expanding and shrinking clusters of T-PLL cells. Left: Heatmap showing the most recurrently enriched HALLMARK pathways²³ and their respective normalized enrichment scores (NES) in a cohort of 7 longitudinal pairs of T-PLL samples. Right: Heatmap of most enriched pathways after inclusion of an additional 4 sample pairs (n=11 total T-PLL pairs). Pathways involved in cell cycle, MYC signaling, and oxidative phosphorylation are among the most recurrently enriched in expanding T-PLL clusters in both cohorts (consistency confirming robustness). (B) Box-whisker-charted mean expression scores of the HALLMARK MYC targets V1²³ gene set comparing the original cohort including 19 T-PLL samples to the extended cohort of 28 T-PLL samples and ten age- and sex-matched healthy donors.²⁴ The overall pattern of a progressive and significant upregulation of MYC target V1 genes from healthy T cells over indolent to active T-PLL cells is preserved and further validated after inclusion of an additional set of 9 T-PLL samples (permutation test, B=10,000). (C) Mean expression scores of the HALLMARK oxidative phosphorylation²³ gene set prior and after expansion of our T-PLL cohort from 19 to 28 T-PLL samples and the inclusion of age and sex-matched healthy donors. Oxidative phosphorylation proves to be significantly upregulated in active stage T-PLL cells while indolent T-PLL cells did not show a significant difference compared to healthy-donor derived T cells (permutation test, B=10,000).

12) The authors show that single-cell analysis results related to MYC targets, oxidative phosphorylation, and glycolysis seem similar to results from previous bulk RNA studies, so it is not evident what novel insights single-cell profiling is providing. Therefore, further deeper analysis of larger single-cell data might provide novel insight into cellular interaction/ communication that is involved in the progression of indolent to active disease. The current findings seem very generalized and associated with the progression of any cancer not only T-PLL.

REPLY: We thank the reviewer for raising this important point. While chromosomal aberrations of chromosome 8q involving the *MYC* locus are indeed established findings in T-PLL,^{25–27} research on the specific role of MYC and its deregulation in T-PLL, to our awareness, has not been conducted. Upregulation of MYC mRNA and protein expression in the context of active-stage T-PLL was first identified by our group in 2018²² and was later validated in smaller cohorts by two other studies.^{28,29} Johansson et al. further observed a downregulation of both mRNA and protein expression of MYC in T-PLL upon *ex vivo* treatment with a CDK9 inhibitor, suggesting MYC to play a central role in cell cycle regulation in T-PLL. In 2021, Tian et al. identified increased levels of H3K4me3 and H3K27ac in a region covering the promoter and gene body of MYC in a genome-wide explorative ChIP-sequencing, potentially contributing to the upregulation of MYC in T-PLL besides copy number amplifications. In line with this, we observed a progressive increase in *MYC* mRNA expression from indolent to active disease even in longitudinal sample pairs without further copy number gains (e.g. pair 2, pair 8, pair 11). This finding provides novel support for epigenetic or transcriptional regulatory mechanisms driving MYC activation in T-PLL. Importantly, studies involving downstream effector functions of MYC in T-PLL have, to our knowledge, not been performed. Furthermore, we are not aware of analyses investigating metabolic deregulations of T-PLL cells. We are certain that our new findings of sub-clonal copy number increases of MYC, the clear pattern of progressive upregulation of MYC mRNA associated with T-PLL tumor progression, and MYC's strong correlation to effector pathways, such as significant upregulation of oxidative phosphorylation, provide significant advances in the understanding of MYC's central role in T-PLL. It also underscores its suitability as a uniform target pathway.

At a more general level, the “novelty” to us is already inherent to the discovery (and our first description) that MYC associated gene signatures become highly prominent upon intra- and cross-individual disease progression of T-PLL. We deliberately focused on this new and main finding based on relevance, rather than limiting the reporting to biologically secondary findings that could have been picked based on their “unknown” nature (“novel”) in the literature, that is biased to active-stage T-PLL and their bulk-analysis. In fact, the unique experimental design of our study provides a relevant value. To date, bulk mRNA sequencing studies in T-PLL have exclusively focused on comparisons between active-stage T-PLL cells and healthy donor T cells. The clinically distinct indolent stage has not previously been captured at the transcriptomic level. Our study is the first to characterize indolent T-PLL cells, allowing us to investigate the emergence and timing of key molecular lesions such as downregulation of the checkpoint molecules *CTLA4* and *LAG3*, which we could establish as early occurring lesions (**Figure 5B**). Further, we identified clonal expansion of ATM-deficient T-PLL cells over disease progression and a strong MYC target gene upregulation (**rebuttal Figure 7C, Figure 3**). This temporal resolution, achieved through longitudinal sampling and intra-individual comparisons, is critical for understanding how T-PLL transitions from an indolent to an aggressive phenotype and cannot be obtained from cross-sectional bulk studies.

In addition, based on the reviewer's suggestion, we expanded our analysis of the non-tumor cell compartments. To our knowledge, this represents the first single-cell characterization of the T-PLL microenvironment. This enabled us to uncover a progressive reduction in cell-cell communication of T-PLL tumor cells and immune cells of the tumor microenvironment. We observed a significant downregulation of genes related to immune signaling and antigen presentation among the non-tumor cells of our samples (e.g. multiple HLA class II genes, **rebuttal Figure 5C**), potentially contributing to immune evasion. These mechanisms would be masked in bulk RNA-seq due to signal averaging and a very small fraction of non-tumor cells.

Finally, the ability to resolve dynamic tumor subpopulations within individual patients allowed us to describe patterns of clonal evolution and shared transcriptional programs and lesions during disease progression, such as subcluster-specific downregulation of *CD45* in 4 out of 11 patients. These insights extend beyond general cancer

hallmarks and contribute to a more refined understanding of T-PLL pathogenesis, providing a framework for identifying stage-specific vulnerabilities and therapeutic opportunities, such as MYC inhibition and modulations of anti-tumor immune responses.

Reviewer #2:

In this study, the authors investigated the potential mechanisms underlying the progression of T-PLL from an indolent phase to an active phase. They identified distinctive features in active-stage T-PLL, including upregulated MYC pathways, increased metabolic activity, downregulation of immune-response pathways and decreased interaction with cells in microenvironment. The study is valuable, well designed and the results are supportive of the conclusions. Couple comments:

REPLY: We thank the reviewer for the thoughtful and constructive comments, and for acknowledging the value and quality of our study. Below, we respond to each point in detail.

1) The authors meticulously detailed the alterations at the RNA level. Have the authors had an opportunity to examine genetic changes at the mutational and karyotyping levels? Specifically, are there any additional mutations and/or cytogenetic changes identified in association with the progression of the disease from an indolent to an active state?

REPLY: We thank the reviewer for this suggestion. In response, we performed whole genome sequencing (WGS) on three matched pairs of indolent and active T-PLL samples (six samples in total). We observed a high concordance between the inferred CNV profiles from the scRNA-seq and those derived from the WGS analysis, particularly for hallmark T-PLL lesions such as *MYC* amplifications and *ATM* deletions (see **rebuttal Figure 3**).

Rebuttal Figure 7

Rebuttal Figure 7. Whole genome sequencing reveals enrichment of *ATM* lesions and *MYC* amplifications in disease progression (A) Total number and predicted type of short variants passing quality filtering in three longitudinal sample pairs as derived from whole genome sequencing (WGS). There was no substantial and homogenous change in mutational burden from indolent to active disease. (B) Bar chart depicting allele frequencies of detected functional coding variants in genes reported to be recurrently mutated in T-PLL.²² All three patients

harbored missense mutations of *ATM* as well as of *JAK/STAT* genes already at the indolent timepoint. **(C)** Dot plot comparing allele frequencies of *ATM* mutations and copy number alterations between indolent and active-stage T-PLL samples. Data were derived from WGS in three longitudinal sample pairs. Mutations of *ATM* with concurring deletions of the remaining allele were detected in all three patients. The variant allele fractions (VAFs) of *ATM* mutations consistently increased in the active timepoint to a near 100% penetrance. **(D)** Copy number alterations of *MYC* in three sequential T-PLL sample pairs analyzed with WGS. The prevalence of *MYC*-amplifications increased from indolent to active disease in all analyzed patients. **(E)** Overview on most relevant emerging genomic lesions during disease progression. Dot plots show the VAFs and predicted copy numbers of affected genes. The top 10 functional coding variants with the highest deltas within their VAFs from indolent to active stage T-PLL are reported for each of the patients. (blue dot: indolent-stage T-PLL; red dot: active-stage T-PLL). Among the top lesions are variants affecting *ATM* (pair_1), *FASTK* (a strong inducer of lymphocyte apoptosis³⁰; pair_2), and *STAT5B* (pair_3).

These analyses revealed several key insights (**rebuttal Figure 7**): Although we did not observe a consistent change in large-scale structural variants or in overall mutation burden during disease progression (**rebuttal Figure 7A**), we detected mutations in genes known to be recurrently affected in T-PLL (**rebuttal Figure 7B**). All three pairs harbored subclonal monoallelic *ATM* mutations at the indolent stage in conjunction with deletion of the remaining allele. The VAFs of these *ATM* mutations consistently increased in the active timepoint to a near 100% penetrance (**rebuttal Figure 7C**, **new Figure 2G**). This suggests that increasingly dysfunctional *ATM* confers a selective advantage during disease progression. Furthermore, the finding of progressive *MYC* amplifications during disease activation, which we had initially inferred from single-cell data, was independently validated by WGS, supporting the biological relevance (**rebuttal Figure 3 and 7D**, **new supplemental Figure 9C**). When investigating emerging variants enriched in active-stage T-PLL, we observed isolated mutations in genes involved in energy metabolism, such as *FASTK* and *SLC2A14*. These may contribute to the metabolic rewiring observed in active T-PLL; however, their functional impact remains unclear and will have to be addressed in future studies (**rebuttal Figure 7E**, **new supplemental Figure 7B**).

2) *JAK/STAT* pathway plays an important role in T-PLL. Is there any change in this pathway (additional mutation involving this pathway or overexpression) during disease progression from indolent to active?

REPLY: We agree with the reviewer that *JAK/STAT* signaling is a critical pathway in T-PLL's biology, as also suggested by our own previous data.^{22,31} To investigate its potential role in disease progression, we examined both transcriptomic and genomic alterations in this pathway using our scRNA-seq and WGS data.

Importantly, we did not observe consistent transcriptional upregulation of *JAK/STAT* pathway associated gene sets (Hallmark IL2 STAT5 Signaling and Hallmark IL6 JAK STAT3 Signaling)²³ during progression from indolent to active disease, suggesting that the pathway's activation status may not shift dramatically at the level of gene expression. At the genomic level, we identified subclonal mutations in *JAK3* and/or *STAT5B* in all three WGS-analyzed patients (**rebuttal Figure 8**, **new supplemental Figure 7C**). In line with our observation regarding *JAK/STAT* pathway gene expression, these mutations did not show a consistent pattern of enrichment during progression. For example, the activating *STAT5B* N642H mutation was present only at the indolent stage in one patient, suggesting that while such lesions may confer a proliferative or survival advantage in experimental systems,³² they do not necessarily drive clonal outgrowth in human T-PLL. This observation is consistent with earlier reports showing that such mutations often remain subclonal and exhibit low VAFs in active-stage T-PLL.³¹ Fittingly, *JAK* inhibitors did not show major clinical efficacy, while newly developed *STAT*-inhibiting compounds, acting more down-stream, have shown promising *ex vivo* efficacy, irrespective of the presence of *JAK* or *STAT* gene mutations.³³

Rebuttal Figure 8

Rebuttal Figure 8. JAK and STAT gene mutations during disease activation

Dot plots displaying the temporal dynamics of identified variants within the JAK/STAT signaling pathway. VAFs are displayed for the indolent and active timepoint. An increase in VAFs was observed for JAK3 p.L875H (pair_1) and STAT5B p.T628S (pair_3) during progression, whereas STAT5B p.N642H (pair_2) and STAT5B p.S635T (pair_3) showed a decline, suggesting reduced clonal relevance over time.

3) In the discussion, there is an opportunity for the authors to further elaborate on their novel findings and explore potential therapeutic implications, particularly considering the poor prognosis associated with T-PLL. The authors mentioned MYC as potential therapeutic target, how about potential targets in metabolic pathways? This study showed downregulation of immune-related signaling pathways. Any potential role of immunotherapy? The findings of tumor-associated monocytes are also interesting, and any potential treatment targeting tumor associated monocytes/macrophage and other cells in the microenvironment?

REPLY: We thank the reviewer for this valuable suggestion. We have expanded the discussion to further explore potential therapeutic implications. We now elaborate more profoundly on the role of MYC as a potential therapeutic target. While MYC activation appears to be an early event, we observe a progressive enhancement of MYC-driven transcriptional programs during disease progression, highlighting MYC and its downstream networks as attractive interventional targets in active T-PLL. Although direct MYC inhibition has historically been considered unfeasible,³⁴ this perception is changing with the development of new MYC inhibitors, one of which has recently entered clinical trials.³⁵ We also discuss the MYC-dependent efficacy of CDK inhibitors in T-PLL,^{28,36} as well as the potential of targeting MYC regulators and effectors.^{29,37} Further, we elaborate on the downregulation of immune signaling pathways, reduced TCR signaling components and MHC expression in T-PLL cells and non-tumor immune cells, as well as the general detachment from the immune microenvironment that we observed in our data. These findings suggest immune evasion mechanisms that could be therapeutically targeted. We discuss why conventional immune checkpoint blockade may be unsuitable for T-PLL and propose alternative immunotherapeutic strategies, including modulation of the CD47/SIRP α axis. We further highlight allogeneic CAR-based cell therapy approaches, which may be particularly effective in the context of an exhausted TME, as they are less susceptible to local immunosuppression.³⁸ These points are now addressed in the revised discussion.

4) The discovery of downregulation of PTPRC (CD45) is intriguing. In my daily practice, I have encountered several cases of T-PLL exhibiting decreased to negative CD45 expression as assessed by flow cytometry.

REPLY: We thank the reviewer for sharing this clinically relevant observation. By expanding our cohort during the revisions of this manuscript, we were able to identify two additional patients with downregulation of PTPRC during tumor progression (pair_8, pair_9). The recurrent downregulation of CD45 (PTPRC), consistent with data from

flow cytometry, provides an interesting feature and should be further interrogated in future research, for example by a systematic assessment of CD45 downregulation in our prospective German T-PLL registry.

5) From Figure 3G, we can see that MYC is active and its expression is increased in indolent phase. The author may want to briefly mention and discuss it.

REPLY: We agree with the reviewer that MYC is already highly expressed at the indolent stage, both at mRNA and protein levels. We have now added a brief statement to clarify this in the text, emphasizing that MYC activity appears to be an early event in T-PLL pathogenesis, but with further amplification and stronger effects on downstream signaling during T-PLL progression.

6) Can the author comment on ATM mutation in T-PLL? Can this be a potential therapeutic target? ATM mutations in T-PLL are very common. In contrast, they are extremely rare in other T-cell lymphomas/leukemias. How is the authors' experience? How often do you see ATM mutations in other T cell neoplasms?

REPLY: As noted above, our WGS data confirmed the presence of *ATM* mutations and deletions in all three assessed patient pairs, with increased clonality observed at the active stage (**rebuttal Figure 7B-C**). These findings support the view that ATM inactivation is a central event in early T-PLL pathogenesis as well as in subsequent progression. *ATM* mutations and deletions are indeed frequent in T-PLL, affecting approximately 81% of patients, while being rare in other T-cell malignancies.²² However, time-resolved data comparing (paired) inactive and active stages of T-PLL had been missing. Our novel data would suggest that ATMs contribution to T-PLL's pathogenesis shows a two-step pattern, with its early role conferred by its initial lesion (mono-allelic CN loss or mutation) followed by its relevance in advanced stages associated with bi-allelic lesions and/or clonal selection. Broader genomic studies report *ATM* mutations in fewer than 10% of PTCL-NOS,^{39,40} with similarly low incidences in T-LGCLL, T-ALL, and AITL.⁴¹ This relative specificity to T-PLL, combined with the clear clonal selection we observed, underscores dysfunctional ATM signaling as a key driver of initial T-PLL leukemogenesis and progression. As we previously demonstrated, ATM deficiency in T-PLL disrupts canonical DNA damage signaling and repair mechanisms, promoting genomic instability and impairing apoptotic responses.²²

This represents a targetable vulnerability. In T-PLL, the functionally hypomorphic ATM is not only depleted in its DNA repair capacity, but is also deficient in its capacity to downstream signal upon (endogeneous or therapeutic) DNA insults. This means that an apoptotic p53 response is insufficiently evoked via ATMs intermediate CHEK2. This inability of conferring a proper ATM-CHEK2 mediated activation of p53 upon DNA damage is thought to confer the notorious chemo-resistance in T-PLL. While *TP53* itself is genomically intact in T-PLL (*TP53* deletions and mutations are rare^{22,42}) p53 is locked in an inactive hypophosphorylated and deacetylated state, bound to its negative regulator MDM2. Therefore, ATM itself is not considered an ideal therapeutic target since its loss-of-function nature can not be reversed pharmacologically; in fact ATM inhibitors showed low in-vitro activity in our hands.²² Consequently, we have been building therapeutic strategies aimed at reactivating p53, either via direct inhibition of MDM2 or by reversing HDAC-mediated epigenetic silencing.⁴² Recent *ex vivo* drug screens have provided functional evidence supporting this rationale in T-PLL.^{36,43}

7) I may miss this information. Did all T-PLL cases in this study carry TCL1 rearrangement? Any cases with t(X;14)?

REPLY: We thank the reviewer for this request for clarification. Not all cases in our cohort carried a TCL1 rearrangement or showed TCL1 overexpression. Specifically, three cases (pair_9, single_indolent_2, single_indolent_3) were TCL1A-negative T-PLL. In one of these cases, we observed a t(X;14) aberration (pair_9). We have now included this information in the revised Supplemental Methods section for clarity.

Reviewer #3; co-reviewed with Reviewer #4:

General comments:

The authors performed single-cell RNA-seq of 7 longitudinally paired and 5 single time-point derived T-PLL samples to study the molecular changes underlying the evolution from indolent to active T-PLL. Based on the fact that T-PLL is a rare cancer, the study of 7 paired cases is of interest, but has limitations. Overall, my feeling is that the study is performed very well, technically there are no concerns and data analysis seems very robust, but due to the low number of cases and the variability from case to case, there are no highly novel conclusions: it is well known that *TCL1A* is implicated in the pathogenesis of T-PLL and also that increased *MYC* expression caused by *MYC* amplification is a common feature of T-PLL.

The analysis of differentially expressed genes (DEGs) in the paired samples is not very informative and seems to indicate that each case has a rather unique set of DEGs.

Since *MYC* is very important in the pathogenesis of T-PLL, it is not surprising that the most uniform shifts of gene expression across all indolent/active sample pairs were detected in GSEA-confirmed functional gene sets associated with cell metabolism and cell-cycle regulation, which are very much controlled by *MYC*, as previously illustrated for T-ALL and other T-cell malignancies.

Interestingly, the authors investigated differential gene expression between T-PLL cell clusters that expanded compared to those that retracted over time and identified in this way a strong enrichment of gene sets related to cell cycle, energy metabolism, and the transcription factor *MYC* in the expanding T-PLL cell clusters. This is an interesting analysis, and one of the strongest parts of the study. However, I am not sure how this was performed as there were no clear clusters shown in the malignant cells based on the analysis of the single-cell data shown in figure 1. How were these clusters identified? I could not find a figure clearly illustrating these clusters.

REPLY:

We appreciate the reviewer's detailed comments and the constructive feedback provided. We acknowledge the limitations posed by sample size. T-PLL is a very rare leukemia with an incidence of approximately 1 per million individuals per year. Beyond that, indolent cases, which represent the focus of our study, constitute only about 20% of all diagnoses. Longitudinally paired, treatment-naïve samples of indolent and active phases are even more scarce. To put this into perspective: since 2008, a dedicated mono-centric collection in Cologne, complemented by a national registry initiative, has facilitated the systematic banking of 496 samples from 252 T-PLL patients. Within this extensive resource, only a very limited number of inactive cases and an even smaller subset of longitudinally matched pairs could be identified. The cohort assembled for the present study therefore represents an exceptional and essentially unique collection, established through more than a decade of sustained and coordinated effort.

At the international level, the initial submission was based on collaborations with major centers and reference collections. For the revision, these efforts were substantially expanded: over the past 18 months, outreach to more than 15 additional research facilities worldwide enabled the inclusion of four further indolent/active pairs and one additional indolent case. As a result, the dataset now comprises 11 T-PLL patients with paired samples, in addition to 5 further inactive and one additional active sample. Together with new analyses and the incorporation of age- and sex-matched healthy controls, this expansion considerably strengthens the robustness, generalizability, and impact of our findings.

Overall, despite inter-patient variability, the initial key findings, such as *MYC* network upregulation, surpassing of metabolic confinement, and shifts in cell-cell communication were consistent and statistically supported across

patients (see **rebuttal Figure 6**). This is very assuring with respect to the robustness of the data and their biological relevance.

Regarding the novelty of MYC-related findings in T-PLL, we agree that MYC alteration is a known feature of T-PLL, primarily based on cytogenetic observations involving chromosome 8q. However, mechanistic insights into MYC's functional role in T-PLL biology remain scarce. In 2018 our group first described MYC mRNA and protein upregulation in active stage T-PLL, which has since been validated in two smaller cohorts.^{22,28,29} Yet, to our knowledge, no previous studies have systematically investigated MYC expression dynamics over disease progression or analyzed MYC-associated metabolic pathway activation in T-PLL. Our data reveal a progressive, clonal upregulation of MYC mRNA, frequent subclonal MYC copy number gains, and a consistent enrichment of MYC target genes in expanding T-PLL subpopulations. Notably, this was accompanied by strong and statistically significant upregulation of pathways such as oxidative phosphorylation and cell cycle regulation - hallmark MYC effector programs. Thus, while MYC is not a novel gene in T-PLL, our study provides the first integrative, functional characterization of MYC's downstream impact on T-PLL subclone fitness and links it to a specific transcriptional and metabolic phenotype. In this respect, the "novelty" to us is already inherent to the discovery (and our first description) that MYC associated gene signatures become highly prominent upon intra- and cross-individual disease progression of T-PLL. We deliberately focused on this new and main finding based on relevance, rather than limiting the reporting to biologically secondary findings that could have been picked as a focus based on their "unknown" nature ("novel") in the literature, which is biased to active-stage T-PLL and their bulk-analysis

Lastly, we thank the reviewer for the interest in the definition of shrinking and expanding clusters. These were derived for each of the T-PLL sample pairs separately. For this, SNN clustering was performed on pairwise integrated T-PLL cell data sets. For each identified cluster, we computed the relative frequency of cells from the indolent and active-stage samples. Clusters were labeled as "shrinking" or "expanding" based on whether their relative abundance significantly differed from the expected distribution of cells across timepoints in a permutation test. Clusters are visualized in **supplemental Figure 5D**.

Below are our responses to each of the specific points raised.

Specific major comments:

1) Figure 1:

On figure 1D/E the single cell data has been integrated removing all patient specific information, but at the same time also removing some of the indolent-active information. Indeed, on figure 1E we see a gradient from indolent to active, but one might expect a clearer separation between the two phases. It would be interesting to see the data before this data integration where the single cells cluster according to the patient and the timepoint. The normal cells of all patients still should cluster together in this plot.

Why does the patient-specific data needs to be removed in this analysis? All subsequent comparisons are on patient level, which means that comparing the indolent with the active samples of the same patient already filters out patient specific information.

REPLY: We thank the reviewer for raising these important questions. In the original version of the manuscript, anchor-based integration was used primarily to facilitate clustering and cell type identification by removing patient-specific technical variation. As commented by the reviewer, we agree that this approach might have inadvertently suggested greater homogeneity among T-PLL samples than is truly present. In response to this and related comments (including those from Reviewer 1), we revised our integration strategy to more accurately reflect the underlying biological heterogeneity among T-PLL samples. Specifically, we removed anchor-based integration between T-PLL samples entirely, as we did not observe substantial batch effects between T-PLL patient datasets. This is supported by our revised analysis (**rebuttal Figure 1**, **supplemental Figure 1B**), which demonstrates that while non-tumor cells from all patients cluster coherently by cell type, T-PLL cells retain their patient-specific expression profiles. This indicates that observed differences among T-PLL cells from different patients are primarily biological rather than technical.

However, we did detect minor, but consistent, batch effects between the non-tumor cell compartment of T-PLL patients and peripheral blood mononuclear cells (PBMCs) from independently processed healthy donors. To address this, we applied a targeted batch correction using the Harmony algorithm with soft parameters ($\theta = 0.1$), limited to this specific comparison. This approach was chosen to reduce technical confounding without overriding meaningful biological differences, especially those between indolent and active T-PLL states. To increase transparency, we now provide tSNE visualizations of merged, uncorrected data alongside batch-corrected embeddings (**supplemental Figure 1B, Figure 1D-G**).

2) In figure 1G the different clusters are annotated. For me it seems weird that the CD4 T-cells cluster more closely to the B-cells than to the CD8 T-cells. Could this be due to the data integration?

REPLY: This observation was indeed related to the earlier use of strongly integrated data, which may have influenced the clustering of cell types. After revising the analysis to avoid over-integration, we now observe that non T-PLL CD4 T cells from both T-PLL patients and healthy individuals cluster closer to CD8 T cells, as expected. We have updated the figure and the figure legend to clarify this point (**new Figure 1D-G, rebuttal Figure 9**).

Rebuttal Figure 9

Rebuttal Figure 9. Predicted cell types in the combined PBMC dataset of T-PLL patients and healthy donors.

t-SNE projection of unsupervised-predicted cell type labels in the PBMC dataset. Colors represent the assigned major cell types. Non-T-PLL cells show homogenous clustering and separation from tumor cells.

3) On line 185 it is stated that a high intra- and inter-sample heterogeneity is observed. This is not clear from this figure as all samples cluster together on the t-SNE plot. In supplementary figure 1B, we see clustering based on the patients, which would indicate that there is not a lot of intra-tumor heterogeneity. Please elaborate.

REPLY: We appreciate this point and agree that the t-SNE representation as displayed in **Figure 1D-G** did not clearly demonstrate intra-tumoral heterogeneity. This was caused by our original approach to use anchor-based integration to facilitate clustering and cell type identification. In the revised version of our manuscript, as outlined above, tSNE representation now clearly depicts inter-patient differences in line with the data presented in **supplemental Figure 1C** (former **supplemental Figure 1B**).

4) In the differential analyses the authors used data on normal cells from an independent study. Why was it not possible to use the normal cells that were sequenced in the current study? How did the authors deal with possible batch effects? Please elaborate.

REPLY:

In the present study, we sequenced samples only from primary T-PLL cases, which naturally contain an extremely small non-malignant T-cell compartment with typically fewer than 150 non-tumor T cells per sample contrasted by over 5,000 malignant T-PLL cells. Comparing these two subsets for differential analysis would lead to strongly biased results. Moreover, these residual non-tumor T cells could themselves be subject to disease-related or microenvironmental influences, including cytokine-mediated signaling, leading to altered transcriptional states. As our dataset demonstrates, the tumor microenvironment in T-PLL exhibits substantial gene expression deregulations affecting antigen presentation and cell-cell interactions. This further limits the validity of using local non-tumor T cells as a control population, at least for questions on tumor-to-normal comparisons.

Therefore, we chose to leverage an external reference consisting of publicly available PBMC datasets from age- and sex-matched healthy donors, processed with comparable single-cell protocols. To address potential batch effects between these datasets, we applied the Harmony algorithm² with conservative parameter settings, allowing us to retain true biological variation while minimizing technical noise. We have revised the Methods section accordingly to clarify this analytical strategy.

5) Figure2:

In figure 2G a GSEA is performed on the pseudotime correlated genes. How were these genes determined? How many genes are included? Is the number large enough to perform GSEA?

Does the pseudotime correlate with the indolent/active timepoints? Maybe add a tSNE/UMAP plot with the pseudotime (in the supplementary)?

What is the added value of this analysis in comparison to the analyses on the indolent-vs-active and the expanding-vs-shrinking? Please explain.

REPLY: We thank the reviewer for this important remark. The pseudotime-correlated genes (gene importance values) were identified using the SCORPIUS pipeline⁴ at default parameters on the full feature matrix (~20,000 genes). Gene set enrichment analysis (GSEA) was performed on the ranked list of gene importance values, ensuring that the analysis remained unbiased and comprehensive. We believe this approach is well-suited for detecting pathway-level changes, and we have updated the Supplemental Methods to clarify the gene selection process.

As for the relationship between pseudotime and disease progression, we confirm that the inferred pseudotime trajectory correlates with the transition from indolent to active disease, as shown in **supplemental Figure 6A**.

We further recognize the reviewer's interest in the added value of this analysis compared to our dichotomous comparisons (indolent vs. active and expanding vs. shrinking clusters). While those analyses capture discrete differences between disease stages or cell populations, pseudotime analysis complements them by modeling the continuous progression of gene expression changes within tumor cells. This dynamic perspective helps to contextualize the intermediate transcriptional states observed in our dataset. However, due to the high degree of patient-specific variation in pseudotime structure and the lack of a meaningful common pseudotime trajectory depicting the overall evolution from indolent to active T-PLL over all samples, we agree that the additional biological insight is limited. To reflect this, we have moved the pairwise pseudotime analysis to the **Supplementary Materials**.

6) Figure 5:

This is well presented, but does not provide a lot of new information as most of the data shown here has been reported previously. Data on CTLA4 were reported in Tian, S. et al. Sci Rep 11, 8318 (2021).

REPLY: We agree with the reviewer that *CTLA4* downregulation has been previously reported.^{22,44} However, we believe our study adds novel insights by delineating the timing of this downregulation during the course of T-PLL leukemogenesis, an aspect that has not been systematically explored before.^{16,36}

Beyond the downregulation of checkpoint molecules shown in **Figure Panel 5B**, we are not aware of prior reports addressing the other findings presented in **Figure 5** in the context of T-PLL. Notably, our observation that TCR activation appears to lose its relevance in advanced stages of T-PLL offers an important refinement of existing disease models, which have emphasized TCR signaling as a key driver of T-PLL pathogenesis.^{22,44} Our data indicate that while indolent T-PLL indeed shows strong TCR pathway activation signatures, these signatures progressively diminish during disease progression.

Furthermore, we identified individual cases displaying downregulation or even loss of TCR signaling components. Of particular interest, we found recurrent downregulation of CD45 (PTPRC) in 4 out of 11 T-PLL cases, which, together with its demonstrated functional relevance for reduced cellular activation upon TCR stimulation (**Figure 5E-G**), expands upon previous single-case observations of CD45 loss in relapsed T-PLL.^{45,46}

7) Figure 6:

In figure 6G/H the cell-cell interactions are investigated. However, only the number of interactions are reported here. It would be interesting to see what kind of interactions are happening (and disappearing). While the changes in cell-cell interactions determined by single-cell RNA-seq analysis are of interest, these remain predicted (and not confirmed) and is not very convincing. Conclusions are limited.

REPLY: We thank the reviewer for this valuable suggestion. In addition to reporting the overall number of inferred cell-cell interactions, we now provide more detailed information regarding the types of interactions that are altered between indolent and active T-PLL in the revised **Figure 6 H-J** and the new **supplemental Figure 13**. Using complementary cell-cell communication frameworks (CellChat⁷ and NicheNet⁸), we identified specific alterations in ligand-receptor interactions and associated downstream effects on transcriptional programs in non-malignant immune cells. Overall, these strongly suggest the pronounced presence of immune-escape mechanisms in active-stage T-PLL.

Specifically, we analyzed potential ligands expressed by T-PLL tumor cells that may contribute to the transcriptomic dynamics observed within the tumor microenvironment during disease activation (**rebuttal Figure 5A**). Among the most consistent and repeatedly identified changes were two genes with well-established relevance in the regulation of the tumor microenvironment, namely upregulation of *ANXA1* (Annexin A1) and downregulation of *CD48*. *ANXA1* is an anti-inflammatory mediator that modulates dendritic cell maturation, monocyte polarization, and T cell activation. Its increased expression in active-stage T-PLL may thus contribute to immunosuppressive reprogramming of surrounding immune cells, promoting a tolerogenic environment. In addition, *CD48*, a co-stimulatory molecule that enhances NK and T cell cytotoxicity via its receptor 2B4 (CD244), was recurrently downregulated. This reduction may impair immune synapse formation and facilitate immune evasion, a mechanism previously described in other hematologic malignancies.

Notably, these ligand changes were paralleled by widespread transcriptional reprogramming in the tumor microenvironment (**rebuttal Figure 5B-C**). We observed downregulation of genes involved in antigen processing and presentation across multiple non-malignant immune subsets, particularly in HLA class II genes within non-tumor T cells, dendritic cells, and B cells. In T-PLL cells, we also noted reduced HLA class I expression. These patterns suggest diminished antigen cross-presentation capacity, which could impair cytotoxic immune responses. In addition, we observed upregulation of *IFI44*, *IFI44L*, and *LY6E* in several immune populations, reflecting an increase in interferon-alpha signaling during disease activation. While traditionally considered tumor-suppressive, recent work has highlighted the role of chronic type I interferon signaling in promoting immunosuppressive circuits, including induction of regulatory cytokines and recruitment of tumor-associated macrophages. Notably, *LY6E* has

been linked to CD8+ T-cell exclusion and resistance to immunotherapy, reinforcing its relevance in shaping an immune-evading microenvironment.

Importantly, we want to emphasize the inherent limitations of interpreting individual ligand-receptor interactions in single-cell RNA-seq datasets, especially in the context of cytokine signaling. Due to the typically low expression levels of many cytokines, compounded by the sparse and shallow nature of scRNA-seq data, such predictions can be highly variable and prone to noise. This challenge is further exacerbated in our study by the relatively small number of non-malignant immune cells and the high degree of inter-patient heterogeneity observed in T-PLL.

Given these limitations, we intentionally focused our primary interpretation on the global pattern of cell-cell communication, as reflected by the total number of interactions and pathway-level changes. We believe this provides a more conservative and robust framework for identifying biologically meaningful trends in the tumor microenvironment. Functional validations of these findings in subsequent efforts are definitely needed.

Minor comments:

8) In figure 5A the heatmap is clustered both on the gene expression and the samples. As this figure wants to indicate a difference between the indolent and the active timepoints (line 328) it would be interesting to see the samples clustered according to their timepoint.

REPLY: As requested by the reviewer we performed clustering of TCR-response transcription factor activities based on the timepoint of samples in addition to the original clustering based on mean enrichment (**rebuttal Figure 10**). While we would generally prefer the unsupervised approach to avoid introducing potential bias, we leave it to the reviewer's discretion which version should be included in the manuscript to best highlight the differences between indolent and active-stage samples.

Rebuttal Figure 10

Rebuttal Figure 10. Signatures of TCR-response transcription factors

Heatmap of scaled TCR-pathway transcription factor signatures in indolent and active-stage T-PLL cells. Enrichment of transcription factor target genes was derived for transcription factors downstream of TCR-signaling using the DoRothEA gene regulatory network. T-PLL samples were arranged in ascending order based on their mean enrichment within indolent and active samples, separately.

9) In figure 6D, there is a character ‘|’ missing in the legend.

REPLY: Thank you for pointing this out. We have corrected this issue in the revised figure.

References:

1. Vu, L. T. *et al.* Single-cell transcriptomics of the immune system in ME/CFS at baseline and following symptom provocation. *Cell Rep Med* **5**, 101373 (2024).
2. Nowotschin, S. *et al.* The emergent landscape of the mouse gut endoderm at single-cell resolution. *Nature* **2019** 569:7756 **569**, 361–367 (2019).
3. Staber, P. B. *et al.* Consensus criteria for diagnosis, staging, and treatment response assessment of T-cell prolymphocytic leukemia. *Blood* **134**, 1132–1143 (2019).
4. Cannoodt, R. *et al.* SCORPIUS improves trajectory inference and identifies novel modules in dendritic cell development. *bioRxiv* 079509 (2016) doi:10.1101/079509.
5. Cao, J. *et al.* The single-cell transcriptional landscape of mammalian organogenesis. *Nature* **2019** 566:7745 **566**, 496–502 (2019).
6. Street, K. *et al.* Slingshot: Cell lineage and pseudotime inference for single-cell transcriptomics. *BMC Genomics* **19**, 1–16 (2018).
7. Jin, S. *et al.* Inference and analysis of cell-cell communication using CellChat. *Nature Communications* **2021** 12:1 **12**, 1–20 (2021).
8. Browaeys, R., Saelens, W. & Saeys, Y. NicheNet: modeling intercellular communication by linking ligands to target genes. *Nat Methods* **17**, 159–162 (2020).
9. Araújo, T. G. *et al.* Annexin A1 as a Regulator of Immune Response in Cancer. *Cells* **10**, 2245 (2021).
10. Perretti, M. & D'Acquisto, F. Annexin A1 and glucocorticoids as effectors of the resolution of inflammation. *Nat Rev Immunol* **9**, 62–70 (2009).
11. McArdel, S. L., Terhorst, C. & Sharpe, A. H. Roles of CD48 in regulating immunity and tolerance. *Clinical Immunology* **164**, 10–20 (2016).
12. Wu, R. *et al.* Epigenetic Modulation of CD48 By NPM-ALK Promotes Immune Evasion in ALK+ ALCL. *Blood* **134**, 1510–1510 (2019).
13. He, M., Yu, J., Chen, S. & Mi, H. A Systematic Immune and Prognostic Analysis of CD48 Interaction with Tumor Microenvironment in Pan-Cancer. *Int J Gen Med* **16**, 5255 (2023).
14. Chiba, M. *et al.* Genome-wide CRISPR screens identify CD48 defining susceptibility to NK cytotoxicity in peripheral T-cell lymphomas. *Blood* **140**, 1951–1963 (2022).
15. Macy, A. M., Herrmann, L. M., Adams, A. C. & Hastings, K. T. Major histocompatibility complex class II in the tumor microenvironment: functions of nonprofessional antigen-presenting cells. *Curr Opin Immunol* **83**, (2023).
16. Ohno, Y. *et al.* IL-6 down-regulates HLA class II expression and IL-12 production of human dendritic cells to impair activation of antigen-specific CD4+ T cells. *Cancer Immunol Immunother* **65**, 193 (2016).
17. Chang, Y. C. *et al.* Epigenetic control of MHC class II expression in tumor-associated macrophages by decoy receptor 3. *Blood* **111**, 5054–5063 (2008).
18. Zhang, X. *et al.* Double-edged effects of interferons on the regulation of cancer-immunity cycle. *Oncoimmunology* **10**, 1929005 (2021).
19. Alhossiny, M. *et al.* Ly6E/K signaling to TGF β promotes breast cancer progression, immune escape, and drug resistance. *Cancer Res* **76**, 3376–3386 (2016).
20. Hailin, L. *et al.* Ly6E on tumor cells impairs anti-tumor T-cell responses: a novel mechanism of tumor-induced immune exclusion. *Cancer Immunol Immunother* **74**, 4 (2024).
21. Braun, T. *et al.* Micro-RNA networks in T-cell prolymphocytic leukemia reflect T-cell activation and shape DNA damage response and survival pathways. *Haematologica* **107**, 187–200 (2022).
22. Schrader, A. *et al.* Actionable perturbations of damage responses by TCL1/ATM and epigenetic lesions form the basis of T-PLL. *Nat Commun* **9**, 697 (2018).
23. Liberzon, A. *et al.* The Molecular Signatures Database (MSigDB) hallmark gene set collection. *Cell Syst* **1**, 417 (2015).
24. Vu, L. T. *et al.* Single-cell transcriptomics of the immune system in ME/CFS at baseline and following symptom provocation. *Cell Rep Med* **5**, (2024).

25. Maljaei, S. H., Brito-Babapulle, V., Hiorns, L. R. & Catovsky, D. Abnormalities of chromosomes 8, 11, 14, and X in T-prolymphocytic leukemia studied by fluorescence in situ hybridization. *Cancer Genet Cytogenet* **103**, 110–116 (1998).
26. Hsi, A. C. *et al.* T-cell prolymphocytic leukemia frequently shows cutaneous involvement and is associated with gains of MYC, loss of ATM, and TCL1A rearrangement. *American Journal of Surgical Pathology* **38**, 1468–1483 (2014).
27. Stengel, A. *et al.* A Comprehensive Cytogenetic and Molecular Genetic Characterization of Patients with T-PLL Revealed Two Distinct Genetic Subgroups and JAK3 Mutations As an Important Prognostic Marker. *Blood* **124**, 1639–1639 (2014).
28. Johansson, P. *et al.* Anti-leukemic effect of CDK9 inhibition in T-cell prolymphocytic leukemia. *Ther Adv Hematol* **11**, 204062072093376 (2020).
29. Tian, S. *et al.* Epigenetic alteration contributes to the transcriptional reprogramming in T-cell prolymphocytic leukemia. *Scientific Reports* **2021 11:1 11**, 1–14 (2021).
30. Tian, Q., Taupin, J. L., Elledge, S., Robertson, M. & Anderson, P. Fas-activated serine/threonine kinase (FAST) phosphorylates TIA-1 during Fas-mediated apoptosis. *J Exp Med* **182**, 865 (1995).
31. Wahnschaffe, L. *et al.* JAK/STAT-Activating Genomic Alterations Are a Hallmark of T-PLL. *Cancers (Basel)* **11**, 1833 (2019).
32. Pham, H. T. T. *et al.* STAT5BN642H is a driver mutation for T cell neoplasia. *J Clin Invest* **128**, 387–401 (2018).
33. Dechow, A. *et al.* Dual STAT3/STAT5 inhibition as a novel treatment strategy in T-prolymphocytic leukemia. *Leukemia* **39**, 1435 (2025).
34. Whitfield, J. R. & Soucek, L. MYC in cancer: from undruggable target to clinical trials. *Nature Reviews Drug Discovery* **2025 1–13** (2025) doi:10.1038/s41573-025-01143-2.
35. Garralda, E. *et al.* MYC targeting by OMO-103 in solid tumors: a phase 1 trial. *Nat Med* **30**, 762–771 (2024).
36. Andersson, E. I. *et al.* Discovery of novel drug sensitivities in T-PLL by high-throughput ex vivo drug testing and mutation profiling. *Leukemia* **32**, 774–787 (2018).
37. Llombart, V. & Mansour, M. R. Therapeutic targeting of ‘undruggable’ MYC. *EBioMedicine* **75**, (2022).
38. Cwynarski, K. *et al.* TRBC1-CAR T cell therapy in peripheral T cell lymphoma: a phase 1/2 trial. *Nat Med* **31**, 137–143 (2025).
39. Watatani, Y. *et al.* Molecular heterogeneity in peripheral T-cell lymphoma, not otherwise specified revealed by comprehensive genetic profiling. *Leukemia* **33**, 2867–2883 (2019).
40. Schatz, J. H. *et al.* Targeted mutational profiling of peripheral T-cell lymphoma not otherwise specified highlights new mechanisms in a heterogeneous pathogenesis. *Leukemia* **29**, 237–241 (2015).
41. Greenplate, A. *et al.* Genomic Profiling of T-Cell Neoplasms Reveals Frequent JAK1 and JAK3 Mutations With Clonal Evasion From Targeted Therapies . *JCO Precis Oncol* **1–16** (2018) doi:10.1200/PO.17.00019.
42. Schrader, A., Braun, T. & Herling, M. The dawn of a new era in treating T-PLL. *Oncotarget* **10**, 626–628 (2019).
43. von Jan, J. *et al.* Optimizing drug combinations for T-PLL: restoring DNA damage and P53-mediated apoptotic responses. *Blood* **144**, 1595–1610 (2024).
44. Oberbeck, S. *et al.* Noncanonical effector functions of the T-memory-like T-PLL cell are shaped by cooperative TCL1A and TCR signaling. *Blood* **136**, 2786–2802 (2020).
45. Thakral, B. & Wang, S. A. T-cell prolymphocytic leukemia negative for surface CD3 and CD45. *Blood* **132**, 111–111 (2018).
46. Chen, X. & Cherian, S. Immunophenotypic Characterization of T-Cell Prolymphocytic Leukemia. *Am J Clin Pathol* **140**, 727–735 (2013).